# Your representations are in the network: composable and parallel adaptation for large scale models

Yonatan Dukler    Alessandro Achille    Hao Yang    Benjamin Bowman*    Varsha Vivek

Luca Zancato    Avinash Ravichandran    Charless Fowlkes    Ashwin Swaminathan

Stefano Soatto

**AWS AI Labs**
{dukler, aachille, haoyng, bowmaben, varshviv, zancato,
ravinash, fowlkec, swashwin, soattos}@amazon.com

## Abstract

We present a framework for transfer learning that efficiently adapts a large base-model by learning lightweight cross-attention modules attached to its intermediate activations. We name our approach InCA (Introspective-Cross-Attention) and show that it can efficiently survey a network's representations and identify strong performing adapter models for a downstream task. During training, InCA enables training numerous adapters efficiently and in parallel, isolated from the frozen base model. On the ViT-L/16 architecture, our experiments show that a single adapter, 1.3% of the full model, is able to reach full fine-tuning accuracy on average across 11 challenging downstream classification tasks. Compared with other forms of parameter-efficient adaptation, the isolated nature of the InCA adaptation is computationally desirable for large-scale models. For instance, we adapt ViT-G/14 (1.8B+ parameters) quickly with 20+ adapters in parallel on a single V100 GPU (76% GPU memory reduction) and exhaustively identify its most useful representations. We further demonstrate how the adapters learned by InCA can be incrementally modified or combined for flexible learning scenarios and our approach achieves state of the art performance on the ImageNet-to-Sketch multi-task benchmark.

## 1   Introduction

Foundation models promise to achieve top performance with minimal adaptation on any downstream task. In the realm of language, the data and the hypothesis spaces are shared, and many tasks can be unified into a homogeneous representation. Visual inference domains, on the other hand, can be highly heterogeneous and possibly antagonistic. For instance, the hypothesis space for pose estimation is geometric, whereas for scene classification it is semantic and even domains that appear homogeneous, such as image classification into the 10 CIFAR classes, can trigger interference in the trained model in the presence of modest perturbations of the image statistics.

Antagonistic domains may interfere within the activations of the latter layers, which cannot be simultaneously minimal and sufficient for all domains. However, information about a dissimilar domain *may be present* in earlier layers, and certainly in the input data which is trivially sufficient

---

*Work conducted while interning at AWS AI Labs.

37th Conference on Neural Information Processing Systems (NeurIPS 2023).

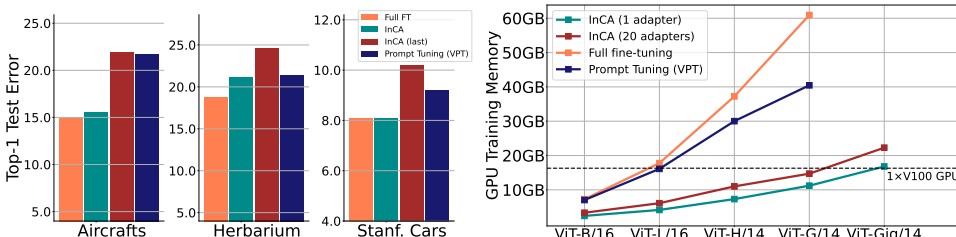

Figure 1: **(Left) Top-1 Test Error** for fine-grained classification transfer learning tasks evaluated with the ViT-L/16 architecture. InCA performs comparable to full fine-tuning on each challenging dataset. **(Right) Max GPU Memory** usage during training for different adaptation approaches and model sizes.

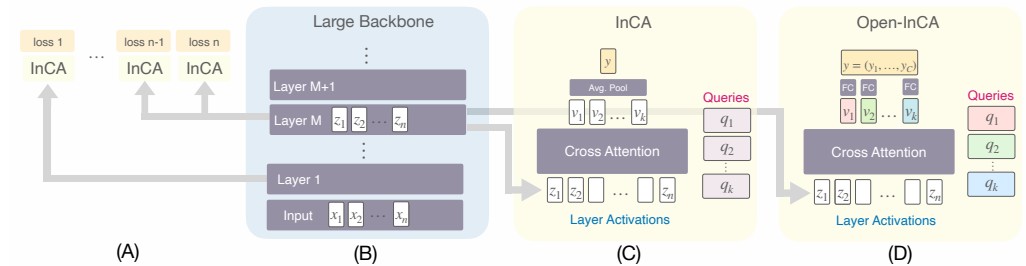

Figure 2: **InCA Adaptation** In (B), intermediate activation maps are extracted from a pretrained backbone during forward pass. Each activation map is passed to a lightweight InCA adapter (shown in (C)) or Open-InCA adapter (shown in (D)) depending on the task settings. In (A), we illustrate how multiple adapters are trained in parallel and independently, and during inference can be combined or used in parallel. In (C), (D) we present a schema of the InCA and Open-InCA adapters; see Sec. 3 for details.

for any task. Indeed, as opposed to just operating with the final model's representations, the typical approach of addressing domain variability in transfer learning is by applying full fine-tuning of the model on new data. By optimizing all of the model's parameters, each of the model representations can be potentially harnessed as a starting point for the adaptation.

While accurate and robust to a variety of domains, full fine-tuning of large-scale models entails sweeping computational and storage costs. Further, the resultant model can only function for its dedicated task, not allowing for shared computation and parallel execution of potential future tasks. To tackle the problem of efficient, versatile, and modular adaptation we introduce Introspective Cross-Attention (InCA). InCA operates on the base-model by attaching isolated shallow adapter modules utilizing any of the activation maps for downstream tasks.

Since modern architectures are deep and so is the variety of possible downstream tasks, iterating over all the possible candidate activations of a model is prohibitively expensive. Instead in InCA we search for useful model representations exhaustively and in parallel by training numerous isolated adapters attached to different activations. When using InCA, each adapter is light and the base-model is fixed and does not require any backpropagation. This makes InCA computationally efficient in terms of GPU memory which is crucial for scaling to large models. Further, the shallow InCA adapter networks simplify the training dynamics as compared with existing approaches which speeds up training considerably and makes the optimization straightforward and robust (see Appendix C).

In detail, during parallel training of InCA, a set of adapters sharing the same architecture are trained simultaneously and independently on a downstream task. Each adapter accepts an assigned activation and does not feed back to the backbone, leaving the backbone execution unaltered during both training and inference (Fig. 2). At inference, the learned adapters may be combined or a best performing adapter can be selected for downstream prediction. The InCA adapter architecture is simple, consisting of a single cross-attention module followed by a linear classifier for the downstream task. Despite the simplicity of the adapter, we observe that a single top-performing adapter trained for a downstream

task is capable of achieving strong performance across a variety of different architectures and pre-trainings when tested on a diverse set of visual domains. Because our approach does not modify the execution of the model or any of the pre-trained layers as done in existing parameter-efficient approaches [28, 34, 27], our method can be automatically applied to *any* architecture without the hassle of re-implementation and architecture specific modifications. In particular, we present results of InCA for ViT [18], SWIN [48], and CNN architectures [49, 74] on a suite of fine-grained recognition tasks.

Since the adapters learned in InCA are shallow and small (1.3% of the parameters on ViT-L/16), the strong performance of a single adapter implies that many pre-trained models *already contain* strong representations for a diverse set of downstream tasks. Instead, previous approaches like linear probing fall short in using the representations, not because the task can not be solved using the existing model, but rather because of not having the right "extraction capacity" of cross-attention; we explore this systematically in Sec. 4 and present a theoretical proof for the advantage of the cross-attention layer in Appendix D. For challenging datasets, using intermediate representations as opposed to only adapting the last layer's representation is key to making InCA a versatile method that closes the gap with full fine-tuning on diverse tasks (See Fig. 1).

A byproduct of the exhaustive approach of InCA adaptation is a signature describing the performance of the internal representations of a network on different downstream tasks. This renders InCA as a powerful tool for understanding the underlying representations of pre-trained models and task space [1] (See Sec. 5, Appendix B). Curiously, we observe that in certain downstream datasets, different pre-trained models hold similar performance signatures, even when the pre-trained models use different architectures or pre-training augmentations.

The isolated training of the adapters means no backpropagation through the pre-trained model is taking place. This significantly reduces the correlation between adaptation cost and the pre-trained model size, which makes it possible to leverage very large architectures even with modest compute resources. Fig. 1 shows that one V100 GPU can train InCA with 40 adapters using an architecture as big as ViT-G/14. In contrast, existing parameter efficient approaches that backpropagate through the architecture exhaust GPU memory with any model larger than ViT-B/16.

Our contributions are summarized as

- We introduce InCA, a method that enables versatile downstream adaptation by surveying the internal network representations of any frozen model with lightweight cross-attention based modules as an alternative to full fine-tuning. On ViT-L/16, InCA matches full fine-tuning performance and reaches within $0.7\%$ accuracy for a SWIN-L backbone on average over 11 diverse downstream domains.
- We demonstrate how the modular adapter architecture of InCA enables flexible learning and inference scenarios and present the Open-InCA adapter. With our approach we unlock powerful pre-trained models for reusable and parallel multi-task inference, and class-incremental learning.
- On the efficiency front, InCA scales to massive scale architectures under typical computation budgets while its implementation can be automatically applied to any new model. For example, training ViT-G/14 using InCA results in 76% GPU memory reduction as compared with full fine-tuning. Further, InCA is easy to optimize and highly parameter efficient (See Appendix C).

The rest of the paper is organized as follows: In Sec. 2 we review related work, and in Sec. 3 we present our approach. We empirically evaluate InCA on a wide set of visual recognition tasks in Sec. 4. Lastly we provide analysis of intermediate representation signatures (Sec. 5) followed by discussion (Sec. 6). Additional results and analysis including Open-InCA are presented in the Appendix.

## 2  Related works

**Transfer learning**  Transfer learning in deep neural networks aims at endowing existing models with new "downstream" knowledge. The de-facto approach for transfer learning in deep learning modifies an existing pre-trained network by applying full, partial or linear fine-tuning on the model [37, 66, 45]. Depending on the the variability between the source and downstream task domains,

different approaches aim at delineating different transfer domains [1, 58, 76], extending coverage of transfer via suitable pre-trainings such as meta-learning [42, 19, 30], or by selecting and combining from a set of expert models [15, 21]. More broadly the field of representation learning focuses on learning transferable representations [38] via different pre-training strategies that can be applied for downstream tasks [11].

**Efficient adaptation methods** In recent times, the top-performing pre-trained model architectures are becoming considerably larger [61, 49], and we arrive at a crossroad as full fine-tuning of such large models is becoming out of reach in many practical settings. Recent works address the storage costs associated with large model transfer by proposing parameter efficient transfer approaches as opposed to storing all of the fine-tuned model parameters [24]. These approaches achieve parameter efficiency by training a subset of existing parameters [6, 45], inserting new weight parameters into existing modules [28, 27, 39, 16] or via additional learnable activations or inputs, known as prompts [34, 65, 43]. Compared with existing work [79, 75], InCA is also parameter efficient, yet we place special emphasis on compute and optimization efficiency, especially in the large-scale model setting. Additional lines of work study learning via selective tuning, enabling multi-domain transfer [70, 22].

**Feature extraction with attention** Self and cross-attention mechanisms aggregate information from a set of feature tokens [69, 41, 44] and create representations based on relative inter-feature importance as computed by the attention mechanism [12, 69]. Self-attention plays a key in the transformer architecture [69] enabling non-local information aggregation. In settings where the number of inputs and desired outputs differ, cross-attention enables flexible aggregation based on a pair of query and key feature sets. When using cross-attention, one can cross-attend between different sets of activations [44, 17] or between a set of activations and learnable latent parameters [32, 9, 80]. In our settings, the adapter architecture applies cross-attention on extracted representations from a pre-trained network, inspired by the cross-attention module of Perceiver [32]. However, we train multiple adapters in parallel and avoid the iterative re-sampling architecture present in their work. More generally, cross-attention layers have been vital in many existing object detection and multi-modal systems that fuse activations [44, 8], or apply cross-attention with learnable latents [3, 32, 77, 9, 80].

**Learning with intermediate features** The re-use of intermediate representations in deep learning is vast and spans from works on interpretability [78], to state of the art approaches in object detection that harness intermediate layers for multi-resolution feature pyramids [47, 68, 20] and segmentation [23, 29]. For ConvNets, the work of [2] studies classification utilizing intermediate network representations and the authors observe a decrease in accuracy when probing earlier layers.

## 3 Method

We introduce InCA, a lightweight and modular transfer learning alternative to full fine-tuning, that avoids backpropagation through the base-model. Let $f(x) = g_n \circ g_{n-1} \circ \ldots g_1(x)$ be a pre-trained feed-forward neural network of $n$ layers, with $g_j(\cdot)$ corresponding to the $j$-th layer of the network. We denote the activation computed by $g_j$ as $f_j(x) = g_j \circ g_{j-1} \circ \ldots g_1(x)$. During network inference, a "forward" computation processes and computes each $f_j(x)$ activation to arrive to the network's final prediction $f(x)$. During standard training, all of the intermediate activations $\{f_1(x), \ldots f_{n-1}(x), f_n(x)\}$ are held in GPU memory and are used to compute gradients to update the model. For large models, this incurs large computational and GPU memory costs [63] which limits using the best and largest available pre-trained models under typical computation budgets.

Instead, we attach a set of isolated "models" to the pre-trained model $f$ at selected activations $f_{j_k}$ and pass them as input to a set of lightweight and shallow networks $h_k(a)$ with separate parameters and losses. With this, we can train a set of heterogeneous adapters $h_k(a)$ in parallel, while computing inference of the pre-trained model $f$ only once during each update (see Fig. 2). For a set of adapters $h_k(a)$ that take as input intermediate activations from $\{f_{j_k}\}$ training follows as:

1. Single inference of $f$ through a data batch $x$ which computes $f(x)$ and selected activations $\{f_{j_k}(x)\}$.[†]

---

[†]We use a callback (though Torch's `register_forward_hook()` or TensorFlow's `get_layer().output`) to cache the activations of the relevant layers which become leafs of the computational graph

2. Calculate the batch predictions and losses for each adapter $h_k$, $\ell_k = \ell(h_k(f_{j_k}(x)), y)$.

3. Computing $\ell_\Sigma = \sum \ell_k$ and applying automatic differentiation then efficiently resolves the gradient and updates of each $h_k$ automatically as desired.

By avoiding backpropagation through the pre-trained $f$ we decouple the majority of the training costs from depending on the size of the base model $f$ and instead the costs correlate with the much smaller adapter set $\{h_k\}$. Below we demonstrate that even a simple cross-attention module for $h_k$ makes the overall adaptation sufficiently expressive yet highly efficient.

**InCA adapter** After extraction of the layer representation $f_k(x)$, we have access to a high-dimensional activation map at our disposal. To predict a target label $\hat{y}$ from the high-dimensional $f_k(x)$, the typical approach is to apply dimension reduction such as averaging (avgpool) or computing maximum values over a subset of the dimensions and then applying a linear classification head:

$$\hat{y} = \text{head} \circ \text{avg-pool} \circ f_m(x).$$

Nonetheless, this simple aggregation approach leads to loss of information which we observe empirically in Sec. 4 and theoretically analyze in Appendix D. Instead, we use a cross-attention module to intelligently aggregate information from the entire large-dimensional activation map $f_k(x)$ into a fixed-dimensional representation based on a number of cross-attention queries. Specifically, for standard downstream adaptation, given an intermediate feature map $\mathbf{z} = [z^1, \ldots, z^T] = f_k(x)$ with $T$ tokens or channels we use the following adapter architecture

$$v_{\text{cross}}(\mathbf{z})_{[1:m]} := \text{cross-attn}_\theta([z^1, \ldots, z^T], [q_1, \ldots, q_m])$$
$$\text{InCA}_\theta(\mathbf{z}) := \text{head}_\theta \circ \text{norm}\left(\text{avg-pool}(v_{\text{cross}}(\mathbf{z})_{[1:m]})\right).$$

Note that the query tokens $[q_1, \ldots q_m]$ are optimized along with $\theta$. The multi-head cross-attention layer outputs $v_{\text{cross}}$ is produced by surveying the feature map $f_k(x)$ with the query tokens $[q_1, \ldots q_m]$. Then, the classification output $\hat{y} = \text{InCA}_\theta(\mathbf{z})$ is obtained through averaging the cross-attention outputs (if $m > 1$) followed by a fully-connected classification head after normalizing with LayerNorm [4]. Based on our experiments, using a single query token $q$ ($m = 1$) achieves strong performance and is computationally efficient and we report results with $m = 1$ unless otherwise stated.

For more flexible inference such as in the settings of continual and class-incremental learning tasks, we present a modular version of InCA that disentangles the representations learned between different classes, which we refer to as "Open-InCA". For a $c$-way classification task, define separate queries $[q_1, \ldots q_c]$ for each class to compute representations separately,

$$[v_{\text{cross}}^1(\mathbf{z}), \ldots, v_{\text{cross}}^c(\mathbf{z})] := \text{cross-attn}_\theta([z^1, \ldots, z^T], [q_1, \ldots, q_c])$$
$$\text{Open-InCA}_\theta(\mathbf{z}) := \text{diag-head}_\theta \circ \text{norm}([v_{\text{cross}}^1(\mathbf{z}), \ldots, v_{\text{cross}}^c(\mathbf{z})])$$

Above, $\text{diag-head}_\theta$ is a linear operator layer that operates on a *matrix* input $[a_1, \ldots, a_c]$ "diagonally". Given a weight parameter $W$, the operator is defined as the column-wise dot product,

$$\text{diag-head}_\theta([a_1, \ldots a_c]) = \left[\langle W_1, a_1 \rangle, \ldots \langle W_c, a_c \rangle\right].$$

**Open-InCA composition** In the Open-InCA adapter architecture, unique queries $[q_1, \ldots, q_c]$ are defined for each class along with $\text{diag-head}_\theta$ that independently processes each coordinate prediction. Both $\text{diag-head}$, LayerNorm and the $\text{cross-attn}$ module in Open-InCA operate on each input $q_i$ independently which separates the representation learned for each class and enables isolating each adapter output coordinate as

$$\text{Open-InCA}(\mathbf{z})_i = \langle W_i, \text{norm}(\text{cross-attn}_\theta([z^1, \ldots, z^T], [q_i])) \rangle.$$

Above, $W_i$ corresponds to the $i$-th column of $\text{diag-head}$ weight. As a result Open-InCA enables class-level modularity with the capabilities of new class insertion, deletion and isolated class updates without regression. For example, deleting class $i$ from the Open-InCA architecture amounts to simply dropping the query and head parameters $q_i$ and $W_i$ for that coordinate. In the setting of class-incremental learning (CIL) different query-head pairs from Open-InCA can be combined together, as long as the parameters of the norm and cross-attn remain the same. In practice, this leads to the notion of training Open-InCA with fixed norm and cross-attention weight parameters, in what we refer to as "*query-only-training*". In query-only-training, the learning of a new class

corresponds to learning just 2, $d$ dimensional parameters per-class and adapter, where $d$ is the token dimension. Nonetheless, when using pre-trained Open-InCA layer parameters, "query-only-training" performs within the accuracy of InCA on many datasets. In Appendix A we compare results of InCA, Open-InCA and query-only-training in class-incremental learning (CIL). In Tab. 6 of the Appendix we observe that even learning just the query and head parameters is capable of harnessing the large dimensional representation maps $f_k(x)$.

**Layer branching candidate selection**  The cross-attention adapters can be applied in parallel over any intermediate layer of the network and we observe that the performance of many tasks hinges on identifying the right intermediate layer to use for that task. When considering intermediate activations $f_k(x)$, we observe that

- Using activations such that $f_j(x)$ is directly computed from a residual connection yields better adapter accuracy. This reflects that network representations are refined through each residual block.

- The middle and later layers of the network provide stronger input representations for the adapter. This is likely since the representations of the early layers do not have discriminative enough features to be used directly for high-level tasks.

**Two-Stage training**  In settings where the base-model forward-propagation during InCA training is too constraining, one may conduct training in two stages. In the first stage, save the activations that serve the input for the adapter for the entire training set, by running the base-model inference for a single epoch. After saving, the second stage proceeds by training the adapters for $T$ epochs with loaded activations. Suppose the per-epoch cost of the pre-trained model forward-propagation is $C_{PT}$ and the per-epoch cost of adapter optimization is $C_A$, then two-stage training reduces the time of training from $O((C_{PT} + C_A) \times T)$ to $O(C_A \times T + C_{PT})$, where $C_{PT} \gg C_A$. With two-stage training, we are able to reduce a 30-epoch adapter training job to 30 seconds for a cached Stanf. Cars dataset (~8,000 training samples). We speculate that further optimization can reduce training costs to "real-time", enabling an array of user-interactive applications.

# 4 Experiments

**Datasets**  In our experiments, we measure the capabilities of InCA on a diverse set of 11 fine-grained datasets consisting of: CUB-200 [73], Aircrafts, [54], Stanford Cars [40], Stanford Dogs [35], Oxford Flowers 102 [56], MIT-67 [60], Oxford Pets [59], Describable Textures (DTD) [13], European Flood [5], FGVC Herbarium [57], and EuroSAT Land Use dataset [26]. In Table 4 we explore InCA in the settings of multi-task learning and evaluate it on the ImageNet-to-Sketch benchmark that is comprised of 5 datasets: WikiArt [64], Oxford Flowers [59], Sketch [71], Stanford Cars [40], and CUB-200 [73].

**Baselines**  For downstream transfer experiments, we compare InCA adaptation to other adaptation protocols. 1) *paragon*: Full fine-tuning is considered as the paragon as it performs well on a diverse set of datasets but incurs steep computational and parameter costs. 2) We compare InCA to other parameter efficient approaches, including 2a) LoRA [28] 2b) Visual Prompt Tuning (VPT) [34] where we apply the top-performing VPT approach, *VPT-Deep*. 2c) BitFit [6] and 2d) AdaLN [45] which is the LayerNorm analogous, AdaBN approach. Note, each approach we name in 2) requires backpropagating through the entire network's activations to update the learnable parameters and leads to a large computational overhead compared with InCA. 3) In addition, we also compare InCA with a suite of computationally efficient approaches that avoid backbone backpropagation like InCA. These include 3a) Linear Probing (LP), 3b) Intermediate Linear Probing (In. LP) which utilizes the same training procedure as InCA but with a LP classifier on the activations. 3c) MLP-3, which is a feed-forward network that consists of probing the base-model with a 3-layer feed-forward network, and 3d) Intermediate MLP-3 (In. MLP-3), the extension of MLP-3 to intermediate layers.

**Training details**  In all of our results (including multi-task settings) we use the same training configuration for InCA. We only change the adapter architecture input layer to automatically match the dimension of the base-model activation map. InCA is robust to hyper-parameters and our training schedule is consistent for all runs. This amounts to 30 training epochs, AdamW optimizer [52] with 2 learning rates, and cosine annealing learning schedule; we provide the full training details

| | Top-1 Test Error, ViT-L/16 | | | | | | | | | | |
|---|---|---|---|---|---|---|---|---|---|---|---|
| Dataset | Full FT | InCA | InCA (last) | In. LP | LP | In. MLP-3 | MLP-3 | VPT [34] | LoRA [28] | AdaLN[45] | BitFit [6] |
| CUB-200 | 9.1 | **8.7** | 9.4 | 16.2 | 16.2 | 13.9 | 13.9 | 10.4 | 12.7 | 15.6 | 15.4 |
| DTD | 18.2 | **17.2** | 18.4 | 18.9 | 20.6 | 17.4 | 20.1 | 21.4 | 19.4 | 22.2 | 21.9 |
| Flood Depth | 18.9 | **17.1** | 19.6 | 17.8 | 22.8 | 17.6 | 20.1 | 19.0 | 19.6 | 18.7 | 18.7 |
| EuroSAT | 1.0 | 1.2 | 1.9 | 2.1 | 3.7 | 1.5 | 2.5 | 1.1 | **0.9** | 1.5 | 1.4 |
| Aircrafts | 14.9 | **15.6** | 21.9 | 50.6 | 67.4 | 36.8 | 47.4 | 21.7 | 16.6 | 28.5 | 27.2 |
| Herbarium | 18.8 | **21.1** | 24.6 | 32.6 | 39.8 | 29.5 | 36.4 | 21.4 | 19.2 | 27.9 | 28.3 |
| MIT-67 | 10.4 | **9.0** | 9.0 | 9.7 | 10.5 | 10.1 | 11.2 | 14.8 | 14.8 | 15.1 | 15.1 |
| Oxford Flowers | 0.6 | **0.3** | 0.4 | 0.6 | 1.1 | 0.5 | 0.7 | 2.2 | 4.0 | 7.0 | 7.2 |
| Oxford Pets | 4.2 | **4.0** | 4.2 | 6.1 | 6.4 | 5.3 | 5.5 | 6.9 | 4.3 | 5.5 | 5.3 |
| Stanf. Cars | 8.1 | **7.7** | 10.2 | 29.2 | 47.2 | 20.8 | 31.4 | 9.2 | 8.4 | 16.0 | 14.7 |
| Stanf. Dogs | 5.9 | 5.4 | 5.8 | 5.3 | 5.3 | 5.7 | 5.7 | 7.3 | 4.3 | 3.8 | **3.7** |
| Mean Top-1 Test Error (Max. gap to Full FT) | 10.0 (0.0) | **9.8 (-2.3)** | 11.5 (-7.0) | 17.2 (-35.7) | 21.9 (-52.5) | 14.5 (-21.9) | 17.7 (-32.5) | 12.3 (-6.8) | 11.3 (-4.4) | 14.7 (-13.6) | 14.4 (-12.3) |
| % Trainable param. | 100% | 1.3% | 1.3% | 0.1% | 0.1% | 2.8% | 2.8% | 0.8% | 2.4% | 0.1% | 0.1% |
| No backbone backprop. | ✗ | ✓ | ✓ | ✓ | ✓ | ✓ | ✓ | ✗ | ✗ | ✗ | ✗ |

Table 1: **Fine-grained Classification Top-1 Test Error (ViT-L/16)** We compare InCA to full fine-tuning (Full FT) along with other adaptation approaches for downstream learning. For each method we summarize the maximum gap in performance compared with the full fine-tuning paragon. In addition, we report the parameter efficiency and whether the method requires backpropagation through the pre-trained model. The minimum error over the columns excluding Full FT is presented in bold.

in Appendix F. While InCA is efficient enough to operate at larger resolutions, we use 224 image resolution unless stated otherwise. Nonetheless, InCA performance improves at 384 resolution while remaining computationally competitive (see Table 2).

**Transfer learning on ViT** In Table 1, we demonstrate the transfer performance of InCA applied to ViT-L/16. For each dataset we train InCA and extract activations at residual layers of the ViT for the last 12 blocks and output layer. For all baselines and our method we use the ViT DeiT pre-training [67] and additionally report ViT-L/16 pre-training results in Appendix Table 8. In the table, we compare InCA to full fine-tuning as well as applying InCA on the last layer and observe that only InCA is capable of achieving good results on challenging datasets such as Aircraft, Stanf. Cars, etc. and closes the maximal gap to full fine-tuning to -2.3%. The second best adaptation approach is LoRA which achieves a maximum gap of -4.4% to full fine-tuning, yet at additional training costs. For a single dataset we can train the InCA modules with 2 learning rates in parallel which corresponds to 26 InCA modules with identical architectures attached to 13 activation maps. In this case, the total training costs of InCA on a single dataset correspond to one base-model run. In Appendix C, we report the hyper-parameter settings and training cost of InCA and current state of the art adaptation method for transformers, VPT [34] which incurs up to $8.7\times$ the training costs of InCA with a large hyper-parameter search (2 vs. 24 settings).

**Transfer learning on SWIN** In Table 3 we present downstream adaptation results for the SWIN-L pre-trained model. InCA adaptation is applied to the 3rd and 4th stages of the network residual activations. Because of the heterogeneous activation dimensions of the hierarchical SWIN architecture,

| | | | | Mean Top-1 Test Error (Max. gap to full FT) | | | | |
|---|---|---|---|---|---|---|---|---|
| Category | Architecture | Pretraining data | Full FT | InCA | InCA (last) | inter. LP | LP | Model size |
| Vanilla Transformer | ViT-B/16 [18] | In21K | 13.0 (0) | 15.9 (-7.6) | 17.5 (-16.4) | 23.9 (-32.4) | 24.3 (-32.4) | 86.5M |
| | ViT-B/16 [44] | ALBEF (CC14M) | 13.8 (0) | 13.5 (-4.2) | 14.8 (-9.3) | 24.7 (-42.6) | 25.8 (-42.6) | 85.9M |
| | ViT-L/16 [67] | In21K (DeiT) | 10.0 (0) | 9.8 (-2.3) | 11.5 (-7) | 17.2 (-35.7) | 21.9 (-52.5) | 304.3M |
| | ViT-L/16 @384 [67] | In21K (DeiT) | -† | 9.2 (-0.6†) | 11.7 (-9.1†) | 17.3 (-38.1†) | 22.0 (-54.4†) | 304.7M |
| | CLIP-ViT-L/14@336 [61] | 400M Im-Text | -† | 9.2 | 10.6 | 19.6 | 21.8 | 304.2M |
| | ViT-H/14 [18] | 2B Im-Text | -† | 9.4 | 10.4 | 14.0 | 15.2 | 632.8M |
| | ViT-G/14 [31] | 2B Im-Text | -† | 9.6 | 10.4 | 15.3 | 16.8 | 1884.9M |
| Hier. Transformer | SWIN-L [48] | In21K | 9.3 (0) | 10 (-3.6) | 12.4 (-9.5) | 15.8 (-31.3) | 18.3 (-40.5) | 196.5M |
| Convolutional | ConvNext-B [49] | In21K | 9.4 (0) | 10.7 (-7.4) | 12.5 (-12.6) | 19.1 (-44.2) | 19.4 (-44.2) | 88.5M |
| | ResNext-101 [74] | IG-3.5B [53] | 11.4 (0) | 12 (-8.7) | 17.3 (-27.1) | 20.1 (-38.8) | 21.3 (-39.7) | 468.5M |

Table 2: **Mean Top-1 Test Error** for transfer learning with a variety of ViT, SWIN, and convolutional networks, including different network scales and pre-training strategies. Averages are reported on the 11 datasets presented in Table 1.† indicates Full FT was avoided due to prohibitive computational costs. For DeiT ViT-L/16 @384 the gap is computed with respect to the 224 pre-training.

the reported adaptation model sizes depend on the activation map used for the selected adapter. InCA achieves the smallest maximum gap to full fine-tuning on SWIN while being computationally efficient. As with ViT-L, in SWIN we observe that challenging datasets require using intermediate activation maps for InCA, closing the maximal gap from (-9.5%) to (-3.6%).

**Evaluating InCA on different pre-trained models**  InCA can be applied to any feed-forward neural network *without* any implementation changes. We simply specify the intermediate layer names and feature tensor-ordering for any new architecture and InCA can be used directly. We note this is in sharp contrast to methods that rely on specific layers such as convolution filters [7, 55] or self-attention [28, 34, 43]. We illustrate the architecture versatility of our method in Table 2. We report the mean and maximum test error gap from full fine-tuning on the 11 fine-grained dataset suite as studied in Table 1. We test different architecture families, which include vanilla vision transformers *i.e.*ViTs, SWIN [48], and modern convolutional networks (ConvNext [49], ResNext [74]). In addition, we test models pre-trained via different strategies including supervised learning [18, 67] and vision-language objectives [44, 61, 31]. We also test InCA at different ViT scales from ViT/B-16 (86M) to ViT/G-14 (1.8B). For InCA adaptation, all model sizes were trained on a single V100 GPU with batch size 32, including for the larger input resolutions runs.

**Multi-task Experiments**  InCA's isolated design is suitable for multi-task inference and a single pre-trained-model can efficiently evaluate a batch of samples on multiple tasks, allowing for "one-to-many" inference. We compare InCA on the ImageNet-to-Sketch multi-task benchmark in Table 4. All methods except BA$^2$ were trained with a ViT-L/16 model and evaluated with the ImageNet-to-Sketch version of each dataset [55]. For BA$^2$ [7], we report the adaptation on a ResNet-50 [25] backbone, as the BA$^2$ approach requires convolutional filters. Overall, InCA is the top performing method reaching near the paragon on the evaluated datasets. Importantly for multi-task, only InCA and LP enable multi-task inference via "computation sharing" of the base model inference.

**Learning efficiency**  Isolating the learning from the base-model means InCA learns shallow neural networks directly on a downstream task. By avoiding deeply backpropagated gradients through the base model, the adapters receive direct signal which improves the optimization dynamics and speed of training. We compare the number of training steps required to train InCA and VPT-Deep and observe that InCA can be optimized in 4.5× fewer epochs than VPT. Here we don't take into account the additional GPU memory costs of optimizing VPT in each step, nor the required hyper-parameter sweeps used in VPT. More detailed efficiency comparison results are given in Appendix C.

| | | | | | Top-1 Test Error, SWIN-L | | | | | |
|---|---|---|---|---|---|---|---|---|---|---|
| Dataset | Full FT | InCA | InCA (last) | In. LP | LP | In. MLP-3 | MLP-3 | LoRA[28] | AdaLN[45] | BitFit[6] |
| CUB-200 | 9.0 | 9.1 | 9.6 | 10.2 | 10.6 | 9.7 | 9.7 | 10.0 | 9.1 | **8.8** |
| DTD | 15.6 | 17.8 | 19.1 | 17.7 | 19.1 | 16.7 | 16.7 | **15.8** | 16.7 | 17.0 |
| Flood Depth | 17.6 | **16.3** | 18.3 | 18.5 | 18.5 | 16.7 | 18.5 | 17.1 | 16.9 | 17.8 |
| EuroSAT | 0.7 | 1.5 | 2.4 | 2.7 | 3.7 | 1.6 | 2.2 | **0.9** | 1.1 | 1.7 |
| Aircrafts | 12.2 | **15.8** | 25.3 | 43.5 | 52.7 | 33.7 | 34.8 | 16.1 | 22.7 | 26.5 |
| Herbarium | 14.9 | **18.2** | 23.0 | 29.2 | 34.0 | 24.9 | 27.6 | 18.4 | 21.2 | 29.7 |
| MIT-67 | 10.5 | 10.1 | 10.1 | 9.9 | 10.3 | 10.2 | 10.2 | 9.6 | 8.9 | **8.5** |
| Oxford Flowers | 0.5 | **0.3** | 0.4 | 0.5 | 0.5 | 0.5 | 0.5 | 0.4 | 0.4 | 0.4 |
| Oxford Pets | 4.6 | **4.7** | 5.5 | 5.0 | 5.5 | 5.5 | 5.5 | 5.2 | 4.8 | 4.9 |
| Stanf. Cars | 7.3 | **8.4** | 15.0 | 29.2 | 39.0 | 22.4 | 26 | 9.6 | 14.2 | 18.4 |
| Stanf. Dogs | 9.1 | 8.1 | 8.1 | **7.1** | 7.1 | 9.8 | 9.8 | 11.3 | 9.1 | 9.0 |
| Mean Top-1 Test Error (Max. gap to Full FT) | 9.3 (0) | **10.0 (-3.6)** | 12.4 (-9.5) | 15.8 (-31.3) | 18.3(-40.5) | 13.8 (-21.5) | 14.7 (-22.6) | 10.4 (-3.9) | 11.4 (-10.5) | 13.0 (-14.8) |
| % Trainable param.$^\S$ | 100% | 3.7% | 3.7 % | 0.1% | 0.1% | 2.8% | 2.8% | 0.8% | 0.1% | 0.1% |
| No backbone backprop. | ✗ | ✓ | ✓ | ✓ | ✓ | ✓ | ✓ | ✗ | ✗ | ✗ |

Table 3: **Fine-grained Classification Top-1 Test Error (SWIN-L)** We compare InCA to full fine-tuning (Full FT) along with other adaptation approaches for downstream learning. For each method we summarize the maximum gap in performance compared with the full fine-tuning paragon. In addition we report the parameter efficiency and whether the method requires backpropagation through the pre-trained model. § For SWIN-L different activations map sizes leads to different % of trainable parameters and we report the maximum for each method. The minimum error over the columns excluding Full FT is in bold.

|  | Top-1 Test Error | | | | | Adaptation Efficiency | | |
|---|---|---|---|---|---|---|---|---|
| Method | Avg. | Flowers | WikiArt | Sketch | Cars | CUB-200 | # of trainable parameters | GPU Memory (training) | Inference Time (for all 5 tasks) |
| Full fine-tuning | 10.5 | 0.6 | **14.7** | 14.4 | 10.8 | 12.2 | 5× | 1× | 5× |
| Linear probing | 29.8 | 10.9 | 37.2 | 29.3 | 44.5 | 27.9 | 0.01× | 0.17× | 1.01× |
| BA$^2$ [7] | 15.9 | 4.3 | 27.7 | 20.7 | 7.9 | 18.8 | 1.03× | 1× | 5× |
| TAPS [70] | 10.4 | 0.6 | 15.8 | **14.0** | 11.1 | 10.4 | 4.12× | 1.23× | 5× |
| SpotTune [22] | 14.3 | 3.7 | 24.2 | 19.8 | **7.6** | 16.0 | 5.27× | 2× | 7.3× |
| **InCA** | **9.8** | **0.3** | 15.4 | 16.8 | 7.7 | **8.8** | 0.06× | 0.51× | 1.13× |

Table 4: **Multitask Efficiency and Top-1 Test Error** on "ImageNet-to-Sketch" benchmark. InCA is the top performing method on average and is parameter efficient. Further, only InCA and linear probing "share computation" of the pre-trained model and enable "one-to-many" inference execution measured in the "Inference Time" column. BA$^2$ is based on ResNet-50 and can not be applied to ViTs. The rest of the methods are based on ViT-L/16.

## 5  Analysis

We analyze the results of InCA adaptation, focusing on the performance signature of different intermediate representations used as input for the adapter and the relation between the top InCA layers with fine-tuning. Further in Appendix D, we provide a theoretical proof motivating the extraction capabilities of cross-attention as it is used in InCA.

**Intermediate representations**  We consider the intermediate representation signature created by evaluating the accuracy of adapters that utilize different layers. In Figure 3, we review the adapter performance applied to different layer representations. Datasets like CUB-200 and Flood-Depth mostly prefer final representations, whereas for datasets like Aircrafts and Stanf. Cars, the best adaptations use earlier representations with decreasing performance towards the last activations. Curiously, we observe consistency in layer affinity for certain datasets while using different pre-trainings for the backbone and even when using different architectures (Appendix Fig. 5).

**InCA and partial-tuning**  In Appendix B we compare InCA with gradually un-freezing the base-model and applying partial fine-tuning on a growing set of layers. We run a set of experiments where we fine-tune a pre-trained model starting at different freezing points, this means we optimize all layers of the network after the freezing point location. For each dataset we construct a "partial tuning curve" where we plot the final test accuracy vs. freezing point (Figure 4). Interestingly, we observe a direct correlation between the layer-location of the top InCA adapter and the point where the partial tuning curve saturates. In particular, the partial tuning test accuracy plateaus (to the tuning of more

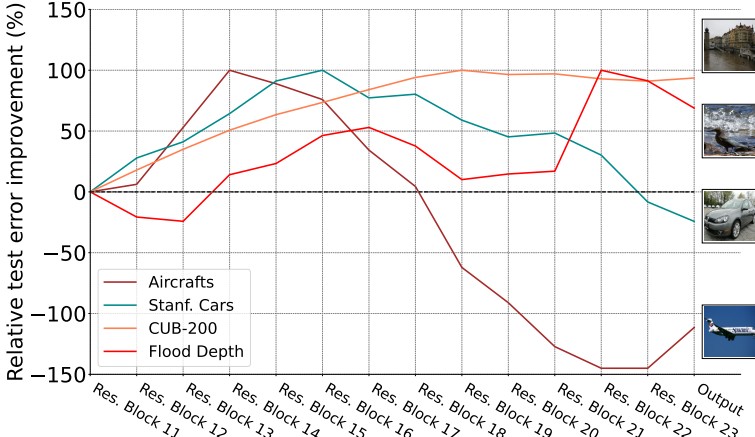

Figure 3: **InCA Layer Performance Signature** Relative test error improvement of InCA adapters attached to different intermediate layers. We evaluate InCA with ViT-L/16 with adapters at each residual block starting from Block 11.

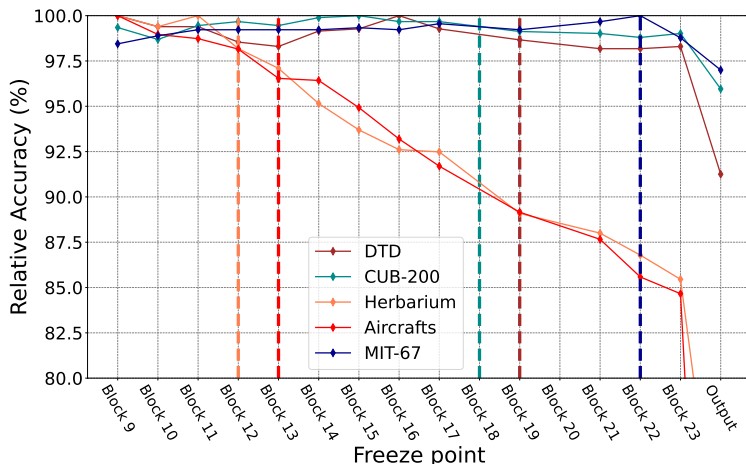

Figure 4: **Partial Fine-tuning vs. InCA** Vertical dashed lines indicate the top InCA layer; curves show final test accuracy for different partial tuning training runs. Each mark indicates a run where all of the pre-trained model parameters are trained up to a "freeze point" in the network's layers. Note partial tuning performance saturates in close proximity to the optimal InCA adapter layer. This is aligned with our hypothesis that full fine-tuning attempts to surface *existing representations already in the network*. In that case, performance improves until the tuning approach unlocks the capacity to utilize an existing relevant representation and performance plateaus afterwards. Note here we refer to output layers, *e.g.*, the adapter at block 19 means the adapter corresponding to the final output of block 19, or the input to block 20.

layers) at around the same layer location as the top performing InCA adapter layer location. Namely, the point of saturation of the partial tuning curve is where partial-tuning is capable of harnessing the representation found by InCA at that layer. This gives further evidence that *"your representations are in the network"* and that fine-tuning surfaces existing representations that can be directly identified by InCA. However, InCA adaptation operates an order of magnitude more efficiently and scales better to large models.

## 6  Discussion

In this paper, we present an efficient and effective alternative to full fine-tuning for transfer learning, closing the gap to full fine-tuning on a diverse set of downstream datasets. InCA has many benefits: it inherently generalizes to different architectures, efficiently scales to massive models, optimizes effectively, and unlocks modular and flexible adaptation applications including multi-task and incremental learning. Further, through the parallel exhaustive search of InCA we are able to better understand the inner representation dynamics of neural networks and construct illuminating "representation signatures" of different models and datasets.

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

# Appendix

Below we provide additional details and results which are not presented in the main manuscript.

## A  Continual Learning with Open-InCA

With the Open-InCA adapter, each class prediction is isolated using a different dedicated query and classifier vector. For continual learning tasks, in addition to running multiple adapters in parallel as presented for multi-task results in Table 4, Open-InCA enables an even more granular composition of adapter sub-tasks. Recall the Open-InCA adapter architecture is defined as

$$[v_{\text{cross}}^1(\mathbf{z}), \dots, v_{\text{cross}}^c(\mathbf{z})] := \text{cross-attn}_\theta([z^1, \dots, z^T], [q_1, \dots, q_c])$$

$$\text{Open-InCA}_\theta(\mathbf{z}) := \text{diag-head}_\theta \circ \text{LN}([v_{\text{cross}}^1(\mathbf{z}), \dots, v_{\text{cross}}^c(\mathbf{z})])$$

with LN denoting LayerNorm. Due to the properties of each operator, each class prediction can be computed separately as

$$\text{Open-InCA}(\mathbf{z})_i = \langle W_i, \text{LN}(\text{cross-attn}_\theta([z^1, \dots, z^T], [q_i]))\rangle.$$

Because of this property we can remove a class prediction or add a new class prediction without any effects on other model predictions (as long as the parameters of cross-attn and LN remain fixed). As presented in Sec. 4 we use "query-only-training" which trains new adapter classes while freezing cross-attn, LN and enabling compatibility between task predictions.

When training with "query-only-training" the softmax function, $\text{softmax}(u) = \frac{\exp(u^k)}{\sum_{i=1}^c \exp(u^i)}$, indirectly injects information of predictions from all classes due the normalization term in the denominator, which means gradients about a particular class $i$ will include information from other classes $j$. Instead, we can achieve complete training separation by using a Sigmoid final activation, $\sigma(u) = \frac{\exp(u)}{\exp(u) + 1}$, and a Binary Cross Entropy (BCE) loss that considers each prediction separately. Clearly in "query-only-training" the adapter representation capacity is reduced, since the cross-attention weights are not trained. We present an experiment evaluating the performance of InCA, Open-InCA and "query-only-training" Open-InCA in Table 6 and observe that despite the isolated and reduced parameter set in query-only-training of Open-InCA, the method is still competitive and outperforms Linear Probing on most datasets.

Next we test Open-InCA for class-incremental learning, for which we consider the Split CIFAR-100 incremental learning benchmark. The Split CIFAR-100 dataset is trained with 10 incremental learning episodes each introducing 10 new classes. As in [10], we present the average episode accuracy and forgetting of "query-only-training" Open-InCa and additional baselines.

In particular we evaluate Open-InCa using a ViT-B/16 along with state of the art methods L2P [72], LwF [46] and EWC [24]. Nonetheless, we do not apply any special routing of our learned episodic models and simply combine their predictions. In contrast L2P is a prompt based approach that, during inference, passes each new sample to an auxiliary classifier to predict its corresponding episode (in this case a 10-way classifier) and the corresponding episode model is up-weighted according to the prediction. We believe that with such an auxiliary classifier Open-InCA performance can significantly improve, nonetheless we observe that Open-InCa can simply leverage a larger model efficiently to achieve state of the art accuracy. We leave routing of samples to different learned sub-models as an interesting avenue for future work.

In addition, Open-InCA has additional benefits as compared to typical class-incremental learning approaches:

- **Flexible incrementation** With Open-InCA different episodes can naturally contain a variable number of classes and episodes can be further decomposed if needed. This is since one can modify the model at the granularity of a single-class predictor via the Open-InCA adapter architecture by introducing or removing additional $q_i$ and $W_i$.

- **Reduced forgetting risk** With Open-InCA the ability of adding new classes without forgetting is built-in into the architecture, as prediction of different classes ensures that the

previous class predictions remain the same (*i.e.*, no logit regression) which reduces catastrophic forgetting.

- **Parameter and computation efficient** The Open-InCA adapter benefits from the InCA approach, which is parameter efficient and computationally efficient during inference (see Table 4) as well as during training (see Fig. 6 for comparison with prompts).

| Method | Average Accuracy (↑) | Forgetting (↓) |
|---|---|---|
| LP-sequential* | 17.7 | 59.1 |
| Full-FT-sequential* | 33.6 | 86.9 |
| EWC [36] | 47.0 | 33.3 |
| LwF [46] | 60.7 | 27.8 |
| L2P [72] | **83.8** | **7.6** |
| Open-InCA (ViT-B/16) | **83.0** | 9.1 |
| Open-InCA (ViT-L/16) | **88.3** | 7.1 |
| Open-InCA (ViT-H/14) | **86.1** | 8.2 |

Table 5: **CIFAR-100 Class-Incremental Learning** Split CIFAR-100 is trained with 10 episodes of 10 classes in the standard CIL evaluation suite [72]. Average accuracy and forgetting is reported over the 10 episodes according with [10]. *Sequential fine-tuning results are taken from [72].

| | | | Top-1 Test Error | | |
|---|---|---|---|---|
| Dataset | InCA | Open-InCA | Query only Open-InCA | In. LP |
| CUB-200 | **9.1** | 9.5 | 12.1 | 16.2 |
| DTD | 17.8 | **17.1** | 19.2 | 18.9 |
| Aircrafts | **15.8** | 18.1 | 38.6 | 50.6 |
| MIT-67 | 10.1 | 9.4 | **9.1** | 9.7 |
| Oxford Flowers | **0.3** | 0.4 | 0.4 | 0.6 |
| Oxford Pets | 4.7 | **4.0** | 5.4 | 6.1 |
| Stanf. Cars | **8.4** | **8.4** | 22.8 | 29.2 |
| Stanf. Dogs | 8.1 | 5.7 | **5.3** | **5.3** |
| Average | 9.3 | **9.1** | 14.1 | 17.1 |

Table 6: **Open-InCA adapter performance** We compare InCA, Open-InCA, "query-training" Open-InCA and Intermediate Linear Probing (In. LP). We observe that Open-InCA is comparable with InCA and that "query-training" significantly out-performs In. LP.

# B   Intermediate Representation Signatures

The parallel training of InCA results in the synthesis of tens of models that can run inference with insignificant per-adapter marginal costs. As a result we have the ability to glean highly useful information about the network's different representations and study the network inner representations effectively. This is especially important for recent non-convolutional based architectures that do not have as many inductive biases explaining some of their behavior. In this section we present results showing information we retrieve from the performance of InCA adapters.

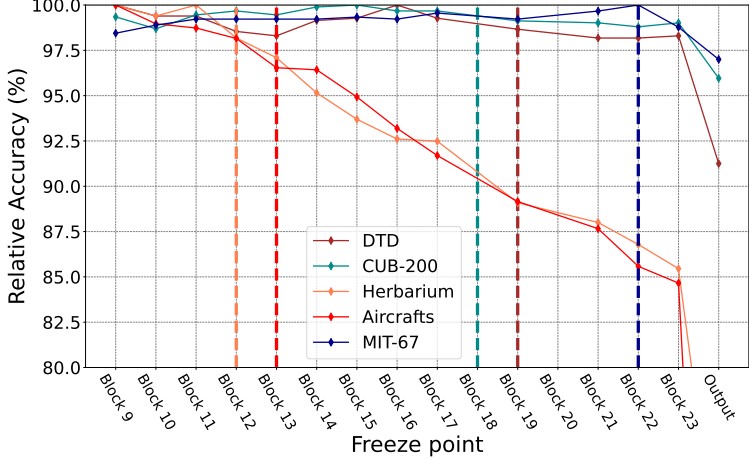

Figure 4: (repeated) **Partial fine-tuning vs. InCA** Vertical dashed lines indicate the top InCA layer; curves show final test accuracy for different partial tuning training runs. Each mark indicates a run where all of the pre-trained model parameters are trained up to a "freeze point" in the network's layers. Note partial tuning performance saturates in close proximity to the optimal InCA adapter layer. This is aligned with our hypothesis that full fine-tuning attempts to surface *existing representations already in the network*. In that case, performance improves until the tuning approach unlocks the capacity to utilize an existing relevant representation and performance plateaus afterwards.

### B.1 Partial fine-tuning and adapter performance

Below we present in detail the experiments discussed in Sec. 5, in particular regarding partial fine-tuning and InCA. The experiments illustrate the relationship between InCA adapters at different layers with partial tuning. We tune the pre-trained model starting from different "freezing points". In particular for neural network $f(x) = g_1 \circ \ldots g_l$, for each freezing point $g_m$ we consider its position $m$ and all of the preceding layers and apply layer freezing to $g_1, \ldots g_{m-1}$ (*i.e.* not updating gradients for those layers). Back-propagation is then only applied for optimizing $g_m, \ldots g_l$ including the network's prediction "head". In Figure 4 we show the dynamics of partial tuning, where we optimize the pre-trained network (ViT-L/16) in different runs with each run having a different freezing point. We compile the final Top-1 Test accuracy of each freezing run to create a partial tuning "curve" for a single dataset. We compare the partial tuning performance curve of each dataset with the corresponding top layer of the InCA adapter trained on that dataset and observe that they are highly aligned, with datasets that prefer later InCA layers plateauing in their test accuracy earlier (at a later freezing point). In particular what we observe is that partial tuning performance plateaus roughly at the same layer where InCA identifies the top adapter representation. This is also the point at which partial fine-tuning is capable of harnessing that representation for the downstream task. Overall this gives further evidence that *"your representations are in the network"* and fine-tuning simply surfaces existing representations that are already identified by InCA. When drawing the vertical lines of the top InCA adapters, we refer to output layers, *e.g.*, the adapter at block 19 means the adapter corresponding to the final output of block 19, or the first input of block 20.

### B.2 Task Layer Affinities

In InCA we select top-performing adapters that "listen" to different intermediate representations of a neural network. In our work we observe that one is able to achieve strong and diverse transfer learning by utilizing intermediate representations, and that for challenging tasks it is often required to use intermediate representations to achieve top results. Indeed the best representation layer for an adapter tends to be highly robust to hyper-parameter variables of the optimization. Even more intriguingly, we find that this representation affinity is preserved across different pre-trainings and even architectures. This is, certain tasks have a strong *"affinity"* to a certain range of representation layers even for different architectural circumstances. The majority of the architectures we consider have some pre-trained component on one of the ImageNet datasets (aside from the CLIP ViT-L/14 model). At the same time, the fact that different architectures give rise to similarly helpful representations gives strong clues about the effect of different architectures as compared with the pre-training task during learning over a large diverse dataset such as ImageNet.

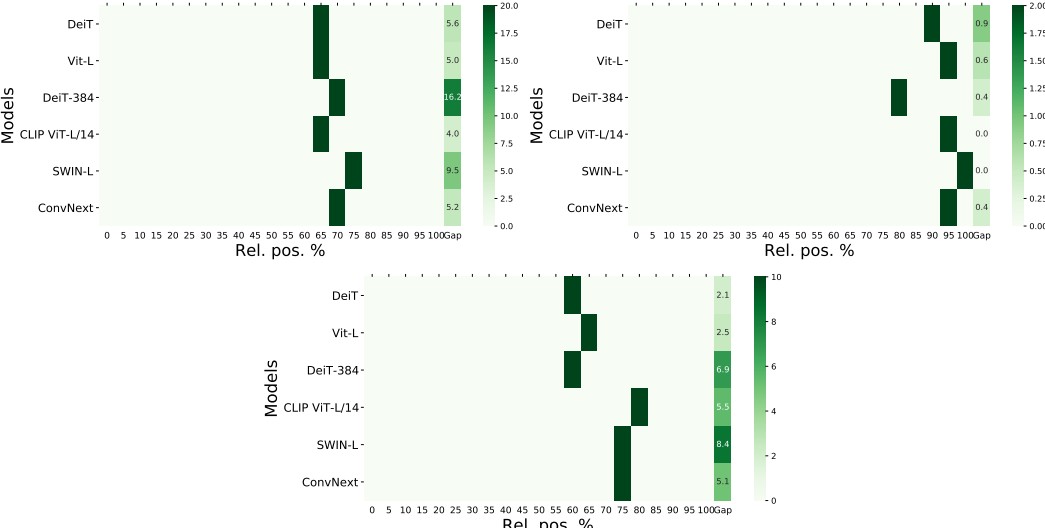

Figure 5: **Best-performing representation for InCA adapter for Aircraft (Top-left) MIT-67 (Top-right) and Stanf. Cars (Bottom).**

In detail, we look at the best-performing InCA adapter for a fixed task on different architectures. The pre-trained models we consider consist of 2 different pre-trainings of the ViT-L/16 architecture (ViT-original and DeiT), the 384-resolution pre-training of DeiT with the resolution adjusted ViT-L/16, CLIP's ViT-L/14 architecture, the SWIN-L architecture, and the convolutional based ConvNext-Base architecture. All of the vanilla ViTs we consider each have 24 residual transformer blocks so that comparing between blocks is directly aligned. SWIN-L and ConvNext follow the "Stage" breakdown of blocks, namely SWIN-L has (2,2,18,2) stage breakdown that conveniently also adds up to 24 blocks (hence aligned in the figure) and lastly, ConvNext follows a (3-3-26-3) + head stage block composition, which we rescale in the figures to fit on the same 24 block range. In addition in the plots we also present the test error gap of each architecture with using the InCA adapter applied on its final block representation. Tasks that prefer earlier layers such as Stanf. Cars and Aircraft have a large gap from the performance of the last layer representation adaptation and such later layers lead to sub-optimal results.

We remark that the work of InCA sheds light on the inner representations learned by neural networks showing in some aspects performance is invariant to the architecture and more based on the pre-training dataset. We leave this topic for further research and find it to be an intriguing topic of study.

## C   Efficiency Results

InCA is highly efficient especially for large models, which is based on the isolated adapter architecture that does not modify the backbone. We delineate the efficiency aspects as follows:

- **Training memory efficiency** The use of a frozen pre-trained model makes the training much more efficient and scalable since not all of the intermediate computations need to be stored as done in standard training or as required by methods that compute gradient information using inner-layers of the network. As soon as *any* intermediate layer requires a gradient, *all* subsequent activations must be held in GPU memory after the forward pass. This means methods like LoRA, FitBit and VPT all require storing of all of the activation maps for all of the layer operations in the network since they update parameters based on gradients from the very early layers in the network.

- **Fast optimization** Unlike typical parameter efficient methods that insert some form of trainable parameters in the network, InCA adapters are trained with "direct gradient" information coming from an isolated loss. Essentially each adapter corresponds to a very shallow neural network trained directly via back-propagation. This makes the training dynamics fast as direct gradient information about the loss easily reaches all of the adapter parameters. On the other hand, to update inserted parameters in the backbone, the gradient information is indirect and needs to be back-propagated through the backbone, with the risk of information loss and making the optimization more challenging, as we and the authors [34] observe regarding prompt tuning.

- **Efficient multi-task inference** As we present in Table 4 the unchanged backbone execution enables efficient and parallel inference efficiency as multiple tasks can be evaluated at once.

### C.1   Computational Efficiency of InCA Compared with VPT

In Table 7 we observe that InCA is an order of magnitude more efficient to train than VPT . For the results in the table we consider the VPT-Deep adaptation method, that is trained with 50 prompt tokens in each layer. We report calculated training times in GPU-hours of a standard Nvidia-T4 GPU using a ViT-L/16 architecture and accuracy numbers based on the datasets of Table 1 with the DeiT pre-training. For larger architectures such as ViT-H/14 ("ViT Huge") the difference in training-time is even more striking, as InCA maintains good per-run training time of 2.5, VPT-Deep requires staggering 55.8 GPU-hours per-run for a single GPU. On ViT-H/14 this is exacerbated as we must reduce the batch-size of VPT significantly to fit training on a common-place single GPU (Nvidia-T4). We measure in terms of training InCA and VPT-Deep for the same number of epochs. This however, is inaccurate as InCA trains an order of magnitude faster on a per epoch basis (see Figure 6).

| Method | Mean Test Err. | Max. Full-FT gap | Training time per run (GPU hrs.) | # Hparam. per dataset | Train time per dataset (GPU hrs.) |
|---|---|---|---|---|---|
| InCA | **10.2** | **2.4** | **2.0** | **2** (parallel) | **4.0 (2.4*)** |
| VPT Deep [34] | 12.3 | 6.8 | 5.8 | 24 | 139.6 |

Table 7: **Computation costs of adaptation** We adapt ViT-L/16 to CUB-200 downstream classification with the same number of training epochs. We evaluate the training and computational costs of a single run and training VPT-Deep and InCA for one training dataset. *Training with 2 learning rates in parallel leads to training time decreasing from 4.0 to 2.4 GPU-hours.

We attribute the difference in training time of InCA to:

1. InCA does not require back-propagation through the whole model which gives $\sim 50\%$ speed improvement alone.

2. InCA is robust to hyper-parameters and we optimize it using just 2 learning rates, compared with the hyper-parameter set of VPT (in our experiments we use 24 hyper-parameter configurations per dataset while using the full configuration presented in [33] takes even longer). In addition, with "one-to-many" training, we train the two hyperparameters of InCA in parallel and report the specific training time in Table 7 denoted by (*) for parallel hyper-parameter training.

3. InCA does not increase the number of propagated tokens in the transformer (*e.g.* in VPT with 100 propagated tokens the attention matrix doubles, from $\sim 4 \times 10^4$ to $\sim 9 \times 10^4$ entries).

### C.2 Optimization dynamics of InCA and VPT

In Fig. 6 we conduct an experiment where we train InCA and state of the art prompting method VPT-Deep [34] for different numbers of epochs and report the final test accuracy. We observe that InCA trains order of magnitude faster than prompting and reaches within $95\%$ relative test accuracy after 3 training epochs.

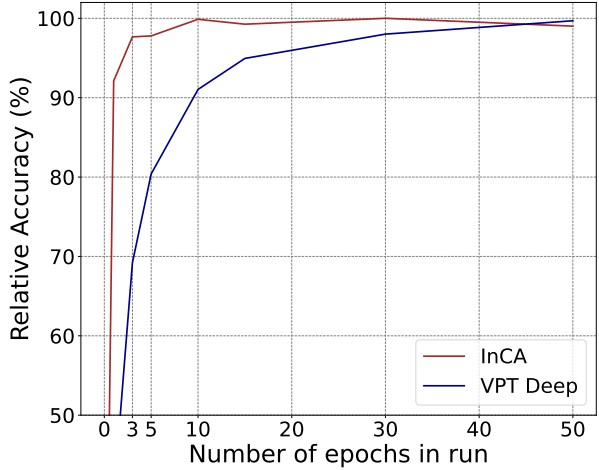

Figure 6: **Optimization Speed** for training InCA and prompt tuning (VPT-Deep) on the Aircrafts dataset. We train each method until completion with varying numbers of epochs and report the relative final test accuracy to 50 epoch training. The shallow adapter architecture and direct gradient signal in InCA makes the training of the adapter an order of magnitude faster (in terms of gradient updates) than prompt tuning approaches. Both methods use batch-size 32 and take the same number of gradient steps in each corresponding run, under the optimal learning rate.

## D  Theoretical Analysis

Empirically, we consistently observe that cross-attention as opposed to a linear or MLP-3 architecture enables InCA to better harness the existing model. We present a theoretical result asserting that using the cross-attention layer for aggregation as opposed to linear averaging, or even full-concatenation followed by a large dimensional linear layer is capable of learning over a strictly broader set of data distributions.

We give the precise statement in Theorem D.1 and intuitively argue that cross-attention with learned queries has the ability to sift through irrelevant pieces of the representation that may be at *variable* positions in different data samples.

Recall in the settings considered thus far, the extracted activation of an image data-point can be viewed as $\mathbf{x}_i \in \mathbb{R}^{d \times T}$ or $T$ tokens *e.g.* $\mathbf{x}_i = [x_i^1, x_i^2 \ldots x_i^T]$, with $x_i^j \in \mathbb{R}^d$. We argue that in many scenarios, task-pertinent information is a property of individual *tokens* (*e.g.* $x_i^j$) within a data-point $\mathbf{x}_i$ and not a property of the overall feature map. We present the theorem below. To this end we define a Token-Separability (TS) notion of a dataset.

**Definition 1** (Token-separable Dataset). *A dataset* $\mathcal{D} = \{(\mathbf{x}_1, y_1), \ldots (\mathbf{x}_n, y_n)\}$ *with* $\mathbf{x}_i = [x_i^1, x_i^2, \ldots x_i^T] \in \mathbb{R}^{d \times T}$ *and* $y_i \in \{-1, 1\}$ *is said to be linearly-token-separable if there exists a scalar* $c > 0$ *and* $w \in \mathbb{R}^d$ *satisfying* $\|w\|_2 = 1$*, such that for each data point* $(\mathbf{x}_i, y_i) \in \mathcal{D}$ *there exists a token* $x_i^{j_i} \in \mathbf{x}_i$ *with*

$$y_i(\langle x_i^{j_i}, w \rangle + b) \geq c. \tag{1}$$

*We define* $(w_\mathcal{D}, b_\mathcal{D})$ *and* $c_0$ *as the maximum margin solution and maximum margin respectively, i.e.* $c_0 = \max_{\{\|w\|=1, b \in \mathbb{R}\}} \min_\mathcal{D} \{y_i(\langle x_i^{j_i}, w \rangle + b)\}$ *for* $\mathcal{D}$ *with* $(w_\mathcal{D}, b_\mathcal{D})$ *corresponding to the selected* $c_0$.

Intuitively, $\mathcal{D}$ is a TS-dataset if each of its data points contain a token that leads to linear separability (the same $w$ shared among all points $\mathbf{x}_i \in \mathcal{D}$ ). One can further distinguish between *aligned*-TS Datasets, where the index $j_i$ of the linearly separating token is consistent among the $n$ data points, or *permutable*-TS where $j$ is dependent on $i$. Further TS datasets can be generalized to $k$-token separable datasets where $k$ tokens are responsible for separability in each $\mathbf{x}_i$, for this theoretical contribution we don't make an assumption on whether the dataset is aligned or permuted, but consider the setting provided by Definition 1 (i.e. not the $k$-separable generalization). We present an analytical statement for the advantage of cross-attn, the theorem is provided for binary classification via a scalar prediction, but can conventionally extend to $C$-class classification. For binary classification we define a prediction via the standard scalar binary aggregator as $\sigma(u) = \text{sign}(\sum_i u_i)$ that converts a vector into a binary prediction.

**Theorem D.1.** *Let* $\mathcal{D}$ *be a binary-class, token-separable dataset with max-margin* $c_\mathcal{D}$ *and max-margin solution* $(w_\mathcal{D}, b_\mathcal{D})$ *consisting of* $n$ *data points. Suppose that* $\mathcal{D}$ *is distributed such that for* $\mathbf{x}_i = (x_i^1, \ldots x_i^T)$ *with* $(\mathbf{x}_i, y_i) \in \mathcal{D}$ *are normalized for separating token* $x_i^{j_i} \in \mathcal{D}$ *and that the rest of the tokens correspond to "noise",* $x_i^k \sim N(0, I/d)$, $\mathbb{E}\|x_i^k\|_2^2 = 1$. *Furthermore assume*

$$c_\mathcal{D} \geq \max\left( \sqrt{\frac{32}{d} \big( \log(1/\delta) + \log(2nT) \big)}, \, 2|b_\mathcal{D}| \right). \tag{2}$$

*Then there exists a cross-attention classifier*

$$f(x; q, \{W\}, b) = \sigma\left( \sum_{l=1}^T \text{cross-attn}(\mathbf{x}, q)_l + b \right) \tag{3}$$

*that separates* $\mathcal{D}$ *with probability at least* $1 - \delta$. *In contrast, every fixed member* $g(\mathbf{x}; w, b) = \sigma\left( \sum_{l=1}^T \mathbf{x}_i \cdot w + b \right)$ *of the linear classifier family will fail to separate* $\mathcal{D}$ *with probability at least* $\frac{1}{\sqrt{2\pi}} \frac{s}{s^2+1} \exp(-s^2/2)$ *where* $s = \frac{\sqrt{d}}{\sqrt{T-1}} c_\mathcal{D}$.

As stated above, the failure probability of the simple linear classifier $g$ depends on $s$ which satisfies $s \sim \sqrt{d/T}$. For existing architectures $d, T$ tend to have a similar order of magnitudes, e.g. for ViT-B/16, $d = 768, T = 196$ which makes the failure probability non-negligible. Before presenting the proof, we make the following observation: in InCA, we use the same cross-attn layer with latent $q$, which we show can be simplified via reparameterization.

**Observation D.2** (Query Collapse Reparameterization). *A single-head cross-attention parameterization with latent $[q]$ is equivalent to the following simplified layer,* cross-attn$(\mathbf{x}, q) = \sum \text{softmax}(q^* \mathbf{x}) \odot \mathbf{W} \mathbf{x}$ *with* $q^* \in \mathbb{R}^d$.

This can be derived by decomposing the attention score which is the input to softmax.

$$a_j = \langle \mathbf{W}_q q, \mathbf{W}_k x^j \rangle = (\mathbf{W}_q q)^\top (\mathbf{W}_k x^j) =$$
$$q^\top \mathbf{W}_q^\top \mathbf{W}_k x^j = (q^*)^\top x^j.$$

Where $q^* = q^\top \mathbf{W}_q^\top \mathbf{W}_k$ and $q^* \in \mathbb{R}^d$, hence the cross-attn layer simplifies, which is used in the proof.

As our proof shows, the cross-attn layer can operate on a large data bandwidth, *e.g.* $\mathbf{x} \in \mathbb{R}^{d \times T}$ while still being selective in finding task specific representations. Empirically we also observe that increasing the number of heads of cross-attn also improves the performance of the InCA . This is in part because it enables the learned latent query parameter $q$ to identify more useful token patterns, and since $q$ is fixed using more heads remain stable (as opposed to when $q$ is a data input). We now present the proof of the claim.

**Proof of Theorem D.1**

*Proof.* The proof of the theorem has two parts A) the positive condition on the cross-attn layer and B) the negative condition on the linear layer (a non-separability probability lower bound). We start with A) and consider separability of positive and negative data samples in turn. First we simplify and write an equivalent cross-attention binary classifier expression for the cross-attn classifier.

**Positive result for the cross-attention model** We consider a "single-head" cross-attention layer and by Observation D.2 we can write the cross-attn layer as follows

$$\text{cross-attn}(\mathbf{x}_i; q, \{\mathbf{W}\}) = \sum_{j=1}^{T} \text{softmax}(\langle x_i^j, q^* \rangle) \cdot \mathbf{W}_v x_i^j. \tag{4}$$

Note that softmax is a function of the entire vector $\{\langle x_i^j, q^* \rangle\}_{j \in [1, T]}$, however we write it in the form above to illustrate the summed terms. For simplicity of notation, we drop the asterisk and write $q^* \in \mathbb{R}^d$ as $q$. Combining cross-attn with the binary aggregator, we have aggregation over the output vector of the cross-attn layer.

$$f(\mathbf{x}_i; q, \{W\}, b)$$
$$= \sigma \left( \sum_{l=1}^{d} \big( \text{cross-attn}(\mathbf{x}_i; q, \{W\}) \big)_l + b \right).$$

Define $S(\mathbf{x}_i, q) \in \mathbb{R}^{1 \times T}$ as the computed softmax argument,

$$S(\mathbf{x}_i, q) = S = \text{softmax}([\langle x_i^1, q \rangle, \dots \langle x_i^T, q \rangle]). \tag{5}$$

Substituting into the classifier, we have

$$f(\mathbf{x}_i; q, \{\mathbf{W}\}, b) = \sigma \left( \sum_{l=1}^{d} \big( \sum_{j=1}^{T} S_j \cdot \mathbf{W}_v x_i^j \big)_l + b \right)$$
$$= \sigma \left( \sum_{j=1}^{T} S_j \sum_{l=1}^{d} (\mathbf{W}_v x_i^j)_l + b \right).$$

Let $u = \sum_{l=1}^{d} (\mathbf{W}_v)_{[l,:]}$ be the sum of the rows of $\mathbf{W}_v$. Note $x_i^j$ can be pulled out from the inner summation to give

$$f(\mathbf{x}; q, u, b) = \sigma \left( \sum_{j=1}^{T} S_j \langle u, x_i^j \rangle + b \right).$$

Thus the cross-attn classifier presented is equivalent to the parameterization above. Next we consider the two terms in the sum, namely $S_j$ and $\langle u, x_i^j \rangle$. We will be deriving their distribution in the case where a data point $(\mathbf{x}_i, y_i)$, has prediction labels $y_i = 1$ and $y_i = -1$ separately. We start with $y_i = 1$ and consider

$$S_k = \frac{\exp(\langle x_i^k, q \rangle)}{\sum_{j=1}^T \exp(\langle x_i^j, q \rangle)}. \tag{6}$$

Take $j_i$ to be the separating token for sample $\mathbf{x}_i$. By the assumption of the theorem for $k \neq j_i$ the tokens correspond to isotropic noise of expected squared norm 1, i.e. $\mathbf{x}_i^k \sim N(0, I/d)$. For a fixed $u \in \mathbb{R}^d$ with $\|u\|_2 = 1$, we take $\eta_k$ to be the distribution of the dot product,

$$\eta_k = \langle u, \mathbf{x}_i^k \rangle = \sum_l (u)_l \cdot (\mathbf{x}_i^k)_l. \tag{7}$$

For $k \neq j_i$ this is a sum of independent Gaussians and each coordinate is distributed as $\sim N(0, \frac{u_l^2}{d})$. As such we have

$$\langle u, \mathbf{x}_i^k \rangle \sim N \left( 0, \sum_l \frac{1}{d} \cdot u_l^2 \right) = N(0, \|u\|_2^2/d)$$
$$= N(0, 1/d)$$

since $\|u\|_2 = 1$. Next we consider $k = j_i$. By the hypothesis we have that

$$y_i(\langle \mathbf{x}_i^{j_i}, w_\mathcal{D} \rangle + b_\mathcal{D}) \geq c_\mathcal{D}.$$

With positive label ($y_i = 1$) this gives $\langle \mathbf{x}_i^{j_i}, w_\mathcal{D} \rangle + b_\mathcal{D} \geq c_\mathcal{D}$. Note that since $c_\mathcal{D} \geq 2|b_\mathcal{D}|$ we have that $\langle \mathbf{x}_i^{j_i}, w_\mathcal{D} \rangle \geq c_\mathcal{D}/2$. Since $w_\mathcal{D}$ is the maximal margin solution, we have $\|w_\mathcal{D}\| = 1$ and $c_\mathcal{D} > 0$. Take $q$ to be of the form $q = t \cdot w_\mathcal{D}$ for $t \in \mathbb{R}^+$ and $u = w_\mathcal{D}$, then

$$t \cdot \eta_{j_i} = \langle x_i^{j_i}, q \rangle = t \langle x_i^{j_i}, w_\mathcal{D} \rangle \geq t \cdot c_\mathcal{D}/2. \tag{8}$$

For $\eta_k$, $k \neq j_i$, separating $t$, we have that $\langle x_i^k, q \rangle = t \cdot \eta_k$. Define $M = \max_{k \neq j_i} (|\eta_k|)$. $M$ is a random variable distributed as the maximum of of $T-1$ i.i.d. Gaussians distributed according to $N(0, 1/d)$. We bound $M$ by investigating an upper bound of the Gaussian CDF. Recall that the moment generating function of a Gaussian random variable $X \sim N(0,1)$ is given by $M_X(r) = \mathbb{E}[e^{rX}] = e^{\frac{1}{2}r^2}$. Then note that for any $s > 0$ we have

$$\mathbb{P}(X \geq r) = \mathbb{P}(e^{sX} \geq e^{sr}) \leq e^{-sr} M(s) = e^{-sr + \frac{1}{2}s^2}$$

where the inequality is an application Markov's inequality. Setting $s = r$ this gives the tail bound

$$\mathbb{P}(X \geq r) \leq \exp(-r^2/2). \tag{9}$$

For our settings with $\eta_k \sim N(0, 1/d)$

$$\mathbb{P}(\eta_k \geq r) \leq \exp(-dr^2/2). \tag{10}$$

For a two-sided bound, by symmetry of the distribution we have

$$\mathbb{P}(|\eta_k| \geq r) \leq 2 \exp(-dr^2/2). \tag{11}$$

Therefore a union bound results in

$$\mathbb{P}(M \geq r) = \mathbb{P} \left( \bigcup_{k \neq j_i} |\eta_k| \geq r \right)$$
$$\leq \sum_{k \neq j_i} \mathbb{P}(|\eta_k| \geq r)$$
$$\leq (T-1) \cdot 2 \exp(-dr^2/2)$$
$$\leq 2T \exp(-dr^2/2).$$

We can bound the bulk of the distribution of $M$ as

$$\mathbb{P}(M < r) \geq 1 - 2T \exp(-dr^2/2). \tag{12}$$

Taking $r = c_{\mathcal{D}}/4$, then with probability at least $1 - 2T \exp(-d(c_{\mathcal{D}}/4)^2/2) = 1 - 2T \exp(-d(c_{\mathcal{D}})^2/32)$ we have

$$M = \max_{k \neq j_i} |\eta_k| < \frac{c_{\mathcal{D}}}{4} \tag{13}$$

and thus

$$\max_{k \neq j_i}(\langle q, x_i^k \rangle) < \frac{tc_{\mathcal{D}}}{4}. \tag{14}$$

With high probability, Eq. (13) holds. This implies that for $j_i$,

$$
\begin{aligned}
S_{j_i} &= \frac{\exp(\langle x_i^{j_i}, q \rangle)}{\sum_{j=1}^{T} \exp(\langle x_i^j, q \rangle)} \\
&= \frac{1}{1 + \sum_{j \neq j_i}^{T} \exp(\langle x_i^j, q \rangle - \langle x_i^{j_i}, q \rangle))} \\
&\geq \frac{1}{1 + \sum_{j \neq j_i}^{T} \exp(\langle x_i^j, q \rangle - tc_{\mathcal{D}}/2))} \\
&\geq \frac{1}{1 + \sum_{j \neq j_i}^{T} \exp(-tc_{\mathcal{D}}/4))} \\
&= \frac{1}{1 + (T-1) \exp(-tc_{\mathcal{D}}/4)} \\
&= 1 - \frac{(T-1) \exp(-tc_{\mathcal{D}}/4)}{1 + (T-1) \exp(-tc_{\mathcal{D}}/4)}.
\end{aligned}
$$

Note that $c_{\mathcal{D}} > 0$ and $T$ is fixed. Nonetheless, the probability bound is independent of $t$ which may take arbitrarily values, e.g. for any $\epsilon > 0$, take $t = 4/c_{\mathcal{D}} \log(T/\epsilon)$, which gives

$$S_{j_i} \geq 1 - \epsilon. \tag{15}$$

Since $S_k \geq 0$ for each $k$ and $\sum_{k=1}^{T} S_k = 1$ we have for $k \neq j_i$,

$$S_k \leq \sum_{j \neq j_i} S_j = 1 - S_{j_i} \leq \epsilon. \tag{16}$$

We consider the classifer prediction

$$f(\mathbf{x}; q, u) = \sigma \left( \sum_{j=1}^{T} S_j \langle u, x_i^j \rangle + b \right) \tag{17}$$

where $b$ is a bias parameter we can choose. Recall $u = w_{\mathcal{D}}$. Focusing on the inside of the sign function

$$
\begin{aligned}
\sum_{j=1}^{T} S_j \langle w_{\mathcal{D}}, x_i^j \rangle &= S_{j_i} \langle w_{\mathcal{D}}, x_i^{j_i} \rangle + \sum_{k \neq j_i} S_k \eta_k \\
&\geq S_{j_i} \langle w_{\mathcal{D}}, x_i^{j_i} \rangle - \sum_{k \neq j_i} S_k \max_{j \neq j_i}(|\eta_j|) \\
&\geq (1 - \epsilon) c_{\mathcal{D}}/2 - \epsilon \cdot c_{\mathcal{D}}/4 \\
&= (1 - (3/2)\epsilon) c_{\mathcal{D}}/2 > \frac{c_{\mathcal{D}}}{4}
\end{aligned}
$$

provided that $\epsilon < 1/3$. If we take $b = -\frac{c_{\mathcal{D}}}{4}$ we have that for $q = t w_{\mathcal{D}}$ and $u = w_{\mathcal{D}}$

$$f(\mathbf{x}_i; q, u, b) = \sigma \left( \sum_{j=1}^{T} S_j \langle w_{\mathcal{D}}, x_i^j \rangle + b \right) = 1 = y_i. \tag{18}$$

Next we address the case where $y_i = -1$. We consider the classifier prediction

$$f(\mathbf{x}; q, u, b) = \sigma\left(\sum_{j=1}^{T} S_j \langle u, x_i^j \rangle + b\right) \tag{19}$$

with $u = w_{\mathcal{D}}$. Again for $k \neq j_i$ we have that

$$\max_{k \neq j_i}(\langle u, x_i^k \rangle) < c_{\mathcal{D}}/4. \tag{20}$$

On the other hand for $j_i$, $y_i = -1$ we have that

$$y_i(\langle x_i^{j_i}, w_{\mathcal{D}} \rangle + b_{\mathcal{D}}) \geq c_{\mathcal{D}}$$
$$\implies \langle x_i^{j_i}, w_{\mathcal{D}} \rangle + b_{\mathcal{D}} \leq -c_{\mathcal{D}}$$
$$\implies \langle x_i^{j_i}, w_{\mathcal{D}} \rangle \leq -c_{\mathcal{D}}/2 < 0$$

where we have used the hypothesis that $c_{\mathcal{D}} \geq 2|b_{\mathcal{D}}|$ in the last line. We consider the term inside the classifier. We note that

$$\sum_{k=1}^{T} S_k \langle u, x_i^k \rangle < \sum_{k \neq j_i}^{T} S_k \langle u, x_i^k \rangle$$
$$< \sum_{k \neq j_i}^{T} S_k \cdot (c_{\mathcal{D}})/4$$
$$\leq c_{\mathcal{D}}/4$$

where in the first inequality we have used the fact that $S_{j_i} \langle u, x_i^{j_i} \rangle < 0$ and in the last inequality we have used the fact that $\sum_{j=1}^{T} S_j = 1$. Therefore again with bias term $b = -c_{\mathcal{D}}/4$ we have that

$$\sum_{j=1}^{T} S_j \langle u, x_i^j \rangle + b < 0 \tag{21}$$

and $f(\mathbf{x}_i; q, u, b) = -1 = y_i$. Thus we have just shown that with probability $1 - 2T \exp(-d(c_{\mathcal{D}})^2/32)$ the model $f(x; q, u, b)$ with $u = w_{\mathcal{D}}, q = tw_{\mathcal{D}}, b = -c_{\mathcal{D}}/4$ gives the correct label for $\mathbf{x}_i$. Taking the union bound over all $n$ points in $\mathcal{D}$ we get with probability at least $1 - 2Tn \exp(-d(c_{\mathcal{D}})^2/32) \geq 1 - \delta$ the model $f(x; q, u, b)$ with $u = w_{\mathcal{D}}, q = tw_{\mathcal{D}}, b = -c_{\mathcal{D}}/4$ separates $\mathcal{D}$.

**Negative result for the linear model** We consider the linear classifier

$$g(\mathbf{x}; w, b) = \sigma\left(b + \sum_{j=1}^{T} \langle w, x^j \rangle\right) \tag{22}$$

where $w \in \mathbb{R}^d$ is restricted to have unit norm $\|w\| = 1$. For an input $\mathbf{x}_i$ under the aggregation, the term inside the sign function simplifies to

$$\sum_{j=1}^{T} \langle w, x_i^j \rangle = \langle w, \sum_{j=1}^{T} x_i^j \rangle.$$

We recall that the $x_i^k$ for $k \neq j_i$ are distributed according to $N(0, \frac{1}{d}I)$. Thus we have that

$$\alpha_i := \sum_{k \neq j_i} x_i^k \sim N\left(0, \frac{T-1}{d}\right). \tag{23}$$

So the problem of classification is equivalent to learning a linear classifier over the separating tokens under the presence of Gaussian noise with distribution $N(0, \frac{T-1}{d})$. Let $i^*$ be the index corresponding to the input $\mathbf{x}_{i^*}$ with smallest margin, i.e.

$$i^* = \operatorname{argmin}_i y_i(\langle w, x_i^{j_i} \rangle + b_{\mathcal{D}}).$$

Then we have that $y_{i^*}(\langle w, x_{i^*}^{j_{i^*}} \rangle + b_\mathcal{D}) \leq c_\mathcal{D}$. We note that any linear classifier $g(\mathbf{x}; w, b)$ with $\|w\|_2 = 1$ will fail to classify $\mathcal{D}$ whenever $y_{i^*}\alpha_{i^*} < -c_\mathcal{D}$. Thus we will lower bound the probability of $\mathbb{P}(y_{i^*}\alpha_{i^*} < -c_\mathcal{D})$. Note for a standard Gaussian random variable $\eta \sim N(0,1)$ as shown in [14] we have for $r > 0$

$$\mathbb{P}(\eta > r) \geq \frac{1}{\sqrt{2\pi}} \frac{r}{r^2 + 1} \exp(-r^2/2). \tag{24}$$

Set $s = \frac{\sqrt{d}}{\sqrt{T-1}} c_\mathcal{D}$. Then by symmetry of the Gaussian distribution the above bound translates into the following bound for $\alpha_{i^*}$

$$\mathbb{P}(y_{i^*}\alpha_{i^*} < -c_\mathcal{D}) \geq \frac{1}{\sqrt{2\pi}} \frac{s}{s^2 + 1} \exp(-s^2/2).$$

It follows that for any $w \in \mathbb{R}^d$ such that $\|w\|_2 = 1$ that the linear classifier $g(\mathbf{x}; w, b)$ incorrectly classifies $\mathcal{D}$ with probability at least

$$\frac{1}{\sqrt{2\pi}} \frac{s}{s^2 + 1} \exp(-s^2/2).$$

This completes the second part of the proof. □

# E  Further Results

We present additional experiments below. In Subsection E.1 we present per-dataset results for additional architectures and a discussion about ensembling InCA is given in Subsection E.2.

## E.1  Per-dataset results for different architectures as presented in Table 2

Table 8 provides per-dataset results that are presented in aggregate in Table 2. Below we present the results for ConvNext-Base and ViT-L/16 (original pre-training) pre-trained models (with the results for ViT-L/16 DeiT and SWIN-L presented in Table 1 and Table 3 respectively).

## E.2  Ensembling learned adapters

Because of "one-to-many" inference of InCA can take a set of independently learned adapters and ensemble them without a marginal increase to the inference cost. We follow non-parametric equal-weight ensembling, by taking the output predictions of two adapters $h_1(x), h_2(x)$ on a sample image $x$. Note that the adapters are computed with their relevant representations via a single forward pass, which makes the execution of $h_1(x)$ and $h_2(x)$ together only incrementally higher than computing just $h_1(x)$. The ensemble is defined as

$$h^*(x) = \frac{h_1(x) + h_2(x)}{2}. \tag{25}$$

Given the large combinatorial selection of $k$ adapters from the $l$ learned adapters we consider the case of ensembling with just two adapter members. After training we evaluate all $m(m-1)/2$ such pairs and compare them with the top performing single layer predictor which we present in Figure 7. In the figure, we illustrate the representations and corresponding adapter pairs that lead to best performance and also present the computed ensemble gain which is the difference between the ensembled model accuracy and the top accuracy of any single adapter.

In addition to improving classification accuracy, ensembling can aid in improving robustness and out of distribution performance which we leave as a future work. Further directions of ensembling include ensembling performance when using adapters of different adapter architectures (e.g. an MLP-3 ensembled with an InCA adapter) or adapters that use representations from different neural networks [21].

## E.3  Ablation on the number of queries

We apply an ablation to see the effects of using a different number of queries in the InCA adapter architecture. In particular, the InCA adapter is written as,

$$v_{\text{cross}}(\mathbf{z})_{[1:m]} := \text{cross-attn}_\theta([z^1, \ldots, z^T], [q_1, \ldots q_m])$$
$$\text{InCA}_\theta(\mathbf{z}) := \text{head}_\theta \circ \text{norm}\,(\text{avg-pool}(v_{\text{cross}}(\mathbf{z})_{[1:m]}).$$

| Top-1 Test Error for ConvNext-B | | | | | |
|---|---|---|---|---|---|
| Dataset | Full fine-tuning | InCA | InCA (last) | Inter. LP | LP |
| CUB-200 | 9.3 | 9.3 | 9.3 | 13.0 | 13.0 |
| DTD | 16.7 | 17.4 | 17.4 | 18.6 | 18.6 |
| Flood Depth | 16.9 | 16.5 | 20.5 | 19.4 | 19.9 |
| EuroSAT | 0.9 | 1.6 | 2.2 | 2.8 | 3.1 |
| Aircrafts | 10.5 | 17.9 | 23.1 | 54.7 | 54.7 |
| Herbarium | 17.0 | 22.7 | 26.4 | 37.4 | 39.5 |
| MIT-67 | 10.9 | 10.3 | 10.7 | 10.2 | 10.4 |
| Oxford Flowers | 0.5 | 0.4 | 0.4 | 0.6 | 0.6 |
| Oxford Pets | 5.2 | 4.6 | 5.6 | 5.9 | 6.0 |
| Stanf. Cars | 6.8 | 9.3 | 14.4 | 39.9 | 39.9 |
| Stanf. Dogs | 8.9 | 7.6 | 7.6 | 7.3 | 7.3 |
| Ave. Top-1 Test Error | 9.4 | 10.7 (-7.4) | 12.8 (-12.6) | 19.7 (-44.2) | 20.0 (-44.2) |

| Top-1 Test Error for ViT-L/16 (ViT pre-training) | | | | | |
|---|---|---|---|---|---|
| Dataset | Full fine-tuning | InCA | InCA (last) | Inter. LP | LP |
| CUB-200 | 11.7 | 10.9 | 10.9 | 12.2 | 12.2 |
| DTD | 18.3 | 18.9 | 20.1 | 19.9 | 20.1 |
| Flood Depth | 20.8 | 18.1 | 18.7 | 18.7 | 18.7 |
| EuroSAT | 0.8 | 1.1 | 1.9 | 2.5 | 3.5 |
| Aircrafts | 20.7 | 23.2 | 28.2 | 44.5 | 46.4 |
| Herbarium | 20.3 | 26.9 | 31.3 | 38.9 | 41.3 |
| MIT-67 | 12.8 | 11.3 | 11.9 | 10.4 | 11.1 |
| Oxford Flowers | 0.6 | 0.3 | 0.4 | 0.3 | 0.4 |
| Oxford Pets | 5.5 | 5.3 | 5.4 | 6.5 | 6.5 |
| Stanf. Cars | 9.3 | 10.9 | 12.9 | 27.6 | 30.2 |
| Stanf. Dogs | 11.0 | 10.4 | 10.4 | 10.1 | 10.1 |
| Ave. Top-1 Test Error | 12.0 | 12.5 (-6.6) | 13.8 (-11.0) | 17.4 (-23.8) | 18.2 (-25.7) |

Table 8: **Per-dataset Adaptation Top-1 Test Error on various architectures** We test transfer learning performance of fine-grained datasets applied to different architectures and pre-trainings including, ViTs, SWIN, and convolutional networks. We report the per-dataset Top-1 test error for the 11 datasets presented in Table 2

.

For $m > 1$ the output of tokens $[q_1, \ldots q_m]$ through the cross-attn layer are averaged, and we test whether using $m > 1$ brings additional representational benefit to each adapter. We present the result in Table 9 and observe that using a different $m$ does not have a consistent effect on the accuracy of the learned adapters, and in our experiments we use $m = 1$ for InCA adapters to be most computationally efficient.

## F   Implementation details

We present the optimization and augmentation details for training InCA, and note we use standardized procedures for augmentation and training (without extensive hyper-parameter optimization) of the different transfer learning methods we evaluate.

**Augmentation**   Unless otherwise specified we train with input image size 224 and standard augmentation practice [62]. In particular, during training we resize to image-size 256 and apply random cropping, for testing we apply resizing and center cropping. For larger image resolutions we maintain the same resize-crop ratio of 0.875.

**Optimization**   For the linear probing and InCA approaches, we train with the AdamW optimizer [50], cosine annealing learning rate scheduler [51] for 30 epochs and with weight decay 1e−4. In each method we sweep over 2 learning rates lr = {1e−4,3e−4}. For full fine-tuning, we also train with AdamW optimizer (weight decay 1e−4), cosine annealing for 30 epochs, but in addition, identify optimal learning rates for each pre-training and architecture separately. We first identify an

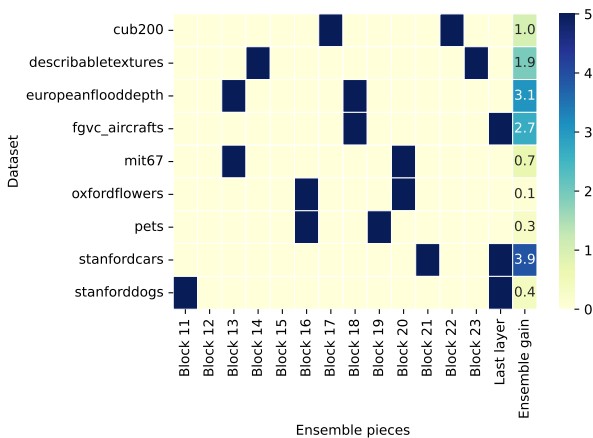

Figure 7: **Optimal Representation pairings** Optimal ensemble pairs of InCA of listeners at different locations of the network; Optimal ensembles can improve over any single layer. ViT-L/16 DeiT pre-training.

| Top-1 Test Error for ViT-L/16 (DeiT pre-training) | | | | |
|---|---|---|---|---|
| | # of InCA queries ($m$) | | | |
| Dataset | 1 | 2 | 4 | 16 |
| CUB-200 | 9.1 | 9.5 | 9.6 | 9.5 |
| DTD | 17.8 | 18.4 | 19.2 | 19.1 |
| Aircrafts | 15.8 | 19.3 | 19.8 | 16.8 |
| MIT-67 | 10.1 | 10.8 | 11.0 | 10.9 |
| Oxford Flowers | 0.3 | 0.3 | 0.3 | 0.4 |
| Oxford Pets | 4.7 | 4.7 | 4.5 | 4.4 |
| Stanf. Cars | 8.4 | 8.7 | 8.8 | 8.2 |
| Stanf. Dogs | 8.1 | 6.3 | 6.3 | 5.9 |

Table 9: **Varying # of queries in the InCA adapter** We run an ablation testing the effect of applying a different number of queries $q_1, \ldots q_m$ and then averaging when using the InCA adapter. We observe that in most cases $m$ does not have a big effect on accuracy and that $m = 1$ has sufficient representation capacity for the adapter.

architecture coarse-range learning rate based on performance on 5 datasets by sweeping over lr = {1e−2,1e−3,1e−4,1e−5,1e−6} followed by a refined sweep with learning rates lr = B, 2B with B being the optimal coarse learning rate.

For the VPT baseline, we follow the details presented in the paper and train with VPT-Deep which was observed to outperforms VPT-Shallow. To train VPT, we use the SGD optimizer with momentum and cosine annealing for 100 epochs. For each dataset we run a sweep on the prompt length {5,20,100}, base learning rate {0.25,0.1,0.05,0.01}, and weight-decay {1e−2,1e−4} for a total of 24 runs with 100-epochs for each dataset. We compare the training cost of InCA and VPT-Deep in Table 7. In general we note that the shallow and small architecture of InCA or linear probing that are separate from the base model makes them straightforward to optimize, compared with adaptation methods that receive back-propagated gradients from a frozen intermediate layer of the network as shown in Fig. 2.

For the LoRA baseline [28] we apply a LoRA modified attention to each block's self-attention layer ($W_k, W_q, W_v$) in ViT based architectures and to each block's WindowAttention for SWIN. For the low rank dimension we sweep over the best value among $d = 5, 10, 50$. For BitFit we follow the discussion in [6] and train all of the bias-parameters in the network in addition to full training of the head. Analogously for [45] we follow their procedure with LayerNorm which includes training each of the LayerNorm parameters ($\gamma, \beta$) for each layer along with training of the head of the pre-trained model. For all of the efficient training methods above we sweep over lr={3e−5,1e−4,3e−4,1e−3} to identify the best learning rate for the dataset.

**Broader Impacts** Our method, InCA enables efficient and modular model adaptation that can be applied to any strong available pre-trained backbone. In that sense, InCA reduces the computational barriers to entry for training and evaluating over a large set of (potentially massive scale) models and optimization settings to identify a model to be used for downstream adaptation. This bridges the gap between cutting edge research in general visual representation learning and specific domain applications, especially since the best performing models are computationally expensive to adapt. Given that InCA operates well on fine-grained visual datasets, this can have positive applications in scientific domains such as medical imaging. In many scientific domains, the available datasets are known to be fine-grained yet also with sparse training data. In addition the ease of use and reduced computational costs associated with downstream adaptation with InCA makes it possible for domain experts without machine learning expertise to use InCA without access to large computational resources. This can enable domain researchers solve their domain problems by leveraging various public pre-trained models to achieve competitive results.

