# OpenReview forum: "Your representations are in the network: composable and parallel adaptation for large scale models"
_NeurIPS.cc/2023/Conference — NeurIPS 2023 poster_

### Official Review · Reviewer_d63D · 2023-07-10

**Soundness:** 3 good
**Presentation:** 4 excellent
**Contribution:** 3 good
**Rating:** 5
**Confidence:** 4

**Summary:**

The paper presents a study on the benefits of training a small cross-attention based adapter instead of performing full fine-tuning of a large VIT model. The authors propose a cross-attention layer they dub InCA that has trainable queries to cross attend to intermediate layers in the large pretrained VIT model and then extract the information by average pooling, layer-norm and Linear. They claim a single query is enough in most cases but also propose a multiple query version they call OpenInCA.

**Strengths:**

* There are strong benefits of having a methodical study of adaptation and I think this paper fits the bill.
* The method is simple, well explained and a priori easy to reproduce and expand.
* The experiments are broad enough to be interesting to the community.
* There are clear computational benefits from the two stage training.

**Weaknesses:**

* The claim in L177 is a bit of a red flag for me. If only one query is enough it may be that the way this is achieved is when the query is very far from the "latent" data and thus the attention weights become flat ie performing average pooling. The nice thing is that it would be an easy and i think useful experiment to check this. Redoing figure 3 with avg pooling but with all the extra bits like the layer-norm that is in the model. I presume the LP doesn't have it ?
* I haven't found any experiments to validate that the class-incremental learning benefits of the Open InCA architecture which is claimed in line 203.

**Questions:**

* Have you tried multiple self-attention layers instead of just one? This would be equivalent to running a small perceiver model [30] and then either average the resulting latent tokens.
* The average pooling has to be done to make sure that the cross attention IS the right architecture here and not a simpler one. It would also add a bit of depth to the paper and answer an obvious question. I think this is required to recommend acceptance.
* Why do you include both tables 1 and 3 ?

**Limitations:**

I think there is no obvious negative impact.

---

> ### Author Rebuttal · Authors · 2023-08-10
>
> We thank the reviewer for the detailed and thoughtful review of our work and the insightful questions. We are pleased that the reviewer appreciates the the methodological experimentation presented in the paper, the generality of the method, and the discussion of two-stage-training. Below we address key questions raised by the reviewer and run the experiments suggested by the reviewer.
>
> >If only one query is enough it may be that the way this is achieved is when the query is very far from the "latent" data and thus the attention weights become flat ie performing average pooling. [An] Experiment to check this... [is] redoing LP with avg pooling but with all the extra bits.
>
> Thank you for the detailed and insightful question regarding the number of queries. The subtlety that can clarify this point is the presence of **multi-head** cross-attention in InCA. With multi-head cross attention, a single query is projected differently for each head. In our case a single query corresponds to 16 different query projections, thereby allowing a single query to surface multiple task specific representations from the activation map.
>
> **Experiment with average pooling**
>
> Following the reviewer’s suggestion, we repeat the adapter experiment using average pooling. Specifically, as you suggest we keep all the extra bits of InCA (e.g. LayerNorm, projection layer of cross attention) and simply replace the learnable cross attention layer with average pooling (but still keep the projection layer for additional capacity). We will refer to this as InLPX for intermediate linear probing “extended”, which now has a layer norm, average pooling, projection layer, and a classifier head. We use InLPX applied to the same activations as InCA in the same exact settings of Table 1 ViT L/16. We report the average test error on the 11 datasets presented in Tab. 1 of the paper in the table below. In short, as the experiment shows, InCA outperforms average pooling which given the experimental settings can be attributed to the relative expressivity of the multi-head cross-attention of the InCA adapter.
>
> **Top-1 Test error over the 11 datasets presented in Tab. 1**
> | Method | Full fine-tuning | InCA | Intermediate linear probing |  InPLX (*New*) |
> | -------- | -------- | -------- | -------- | -------- |
> | Average | 10.0  | 9.8 | 17.2 | 14.1 |
> | Maximum gap to full FT | 0 | 2.3 | 35.7 | 26.4 |
>
> As you suggest we observe that adding the additional bits of InCA to intermediate linear probing does lead to an improvement, namely from 17.2 → 14.1 average test error. **However**, even by repeating the entire procedure aside from cross attention as in InCA we still observe that InLPX is still has a maximum gap of 26.4 points as compared with InCA that has 2.3 point gap (>10x larger maximum gap), and has 44% worse test error relative to InCA.  Given the experimental design of making InLPX match InCA on all aspects except cross attention, the gain can be directly attributed to the cross attention layer.
>
> **Detailed explanation on expressivity**
>
> We point out 2 key factors when looking at cross attention vs. average pooling
> 1. **Multi-head**: The InCA adapter uses a multi-head cross attention module which has 16 heads by default (follows the number of heads in the pre-trained model). In this example, the single query is equivalent to 16 distinct smaller dimensional queries applied on the activation map, enabling diverse querying of task-relevant representations.
>
> 2. **Instance-based**: Further another aspect we believe is important in the cross attention layer is that even if the queries are fixed, the aggregation of the activation map will not be constant between images, this is because the computed keys of the activation map are going to depend on the particular input instance which will further result in dynamic aggregation.
>
> Lastly note Appendix D is concerned with analytically describing the benefits of cross attention and can be relevant to this discussion.
>
> >I haven't found any experiments to validate that the class-incremental learning benefits of the Open InCA architecture which is claimed in line 203.
>
> The experiments for Open-InCA are presented in Appendix A. We provide a quick summary of the results below. Recall in Open-InCA, we learn a query specific to each class separately, as well as class specific heads (diag-head). This completely disentangles the learning of additional classes from existing classes which enables extremely flexible class incremental learning and class forgetting. Out of the box, Open-InCA achieves competitive results on the Split-CiFAR100 Class Incremental Learning benchmark without task specific architectural changes (e.g. a split routing classifier) used by state of the art methods.
>
> >have you tried multiple self-attention layers instead of just one? This would be equivalent to running a small perceiver model [30] and then either average the resulting latent tokens.
>
> We have not trained InCA adapters with access to multiple activation maps, this is an interesting future research. However, we observe that a *single adapter* utilizing a *single relevant activation map* is already capable of outperforming all other efficient adaptation methods, achieving within the same accuracy on average as full fine-tuning for ViT-L/16. We also point the reviewer to Appendix E2 that presents a preliminary ensembling study of InCA adapters.
> >Why do you include both tables 1 and 3 ?
>
> Table 1 and Table 3 of ViT and SWIN architectures are included to demonstrate the generality and diversity of InCA on non-vanilla transformer architectures.
>
> **Conclusion** We thank the reviewer again for the thoughtful review of this work. We hope that by addressing the main concerns raised, specifically 1) the additional details on Open-InCA experiments (presented in Appendix A) and 2) clearing up the query question and repeating the experiment with average pooling, the reviewer considers raising their recommendation.

---

> > ### Comment · Reviewer_d63D · 2023-08-18
> > **comment**
> >
> > I thank the authors for their additional work and comments. I think this makes the paper more easy to justify in my mind. i will thus change my ranting to borderline accept.

---

> > > ### Author Response · Authors · 2023-08-19
> > > **Response to reviewer**
> > >
> > > We thank the reviewer for taking the time to consider our rebuttal response and change their rating accordingly.

---

### Official Review · Reviewer_XAoM · 2023-07-12

**Soundness:** 3 good
**Presentation:** 2 fair
**Contribution:** 2 fair
**Rating:** 6
**Confidence:** 4

**Summary:**

- This paper proposes a new way to adapt a pretrained deep neural network for downstream tasks called InCA. InCA does not modify the intermediary representations of the pretrained network and thus doesn’t require backproping through it, which makes it memory- and compute-efficient. To use InCA, one first heuristically identifies a few layers, the activations of which are processed through cross-attention with trainable query tokens.
- The cross-attention setup is more expressive than linear maps, which is supported by empirical and theoretical arguments. It also allows the addition of new classes through the injection of new query tokens.
- Finally, because the forward pass of the original network is not at all modified, one can cache the relevant activations of the entire dataset once and apply InCA without calling the original model at all. It is also possible to adapt to many tasks in parallel for the same reason.
- Experiments on many image datasets show that InCA performs competitively with finetuning and other efficient adaptation methods.

**Strengths:**

- This work includes a broad range of baselines in their experiments.
- The proposal is simple yet appears to be effective. The stated benefits are technically sound.

**Weaknesses:**

- The paper “[presents] a framework for transfer learning that efficiently adapts a large base model by learning lightweight cross-attention modules attached to its intermediate activations.” However, the experiments are limited to mostly computer vision classification tasks. Some of the baseline approaches, such as LoRA and BitFit, are extensively used for language understanding, language generation, and image diffusion. The result of this paper will be much more convincing if experiments in these domains are included, especially if InCA can outperform existing baselines.
- A significant portion of the benefit of InCA comes from using a subset of the layer activations. However, this important choice (as described in B.2) is done heuristically according to Section 3. The paper can benefit from more clarity on how to choose such layers. For example, how are these layers chosen for the experiments?
- The clarity of the writing can be improved. For example, it was not clear during my first pass what the dimension of z is and what T is. The description for diag-head is confusing: it appears to be a simple matrix multiplication between [a_1, …, a_c] and W. Section 3 overall can be better organized by heuristics, methodology, and benefits.
- The “signature” phenomenon is interesting, but the given analysis doesn’t provide much insight. It would be great to hear more about what we can learn from these “signatures,” especially given that they are presented as one of the key advantages of InCA in the conclusion.

**Questions:**

- How competitive is InCA in other domains where many of the compared baselines, e.g., LoRA and BitFit, dominate? E.g., language understanding, language generation, diffusion, etc.
- How is the subset of layers chosen for the experiments? Could a similar selection process benefit other baselines?

---

> ### Author Rebuttal · Authors · 2023-08-10
>
> We thank the reviewer for the thorough review of our work and appreciate that they found InCA a simple and comprehensive approach with broad range of baselines and sound benefits. below we address the main points.
>
> >A significant portion of the benefit of InCA comes from using a subset of the layer activations... how are these layers chosen for the experiments?
>
> We would like to clarify our layer selection process — we do not heuristically select the layers for each experiment, rather we use the parallel training property of InCA (Lines 148-157) to **exhaustively train all layers in parallel**. We test the performance of each adapter using cross validation and finally report the best performing layer (retrained on all the training set). This is one of the key motivations of InCA, we wanted to find a way to exhaustively analyze each layer. Done sequentially, this is very expensive, instead InCA’s parallel training becomes an effective solution since we are capable of training 40+ adapters in parallel on a single GPU.
>
> In Sec. 3, we note that when running the experiments *we patently* observe that the best performing layers are from activations with residual connections and tend to be from the later half of the network. Therefore we use that simple criteria for the adapter set, however we can, and have trained InCA attached to all of the network’s blocks.
>
> > Could a similar selection process benefit other baselines?
>
> Yes, in Tab. 1 and 2 we report the In-LP and In-MLP-3 adaptation results (with “In” refers to intermediate). This corresponds to applying InCA’s parallel training procedure with intermediate representations for linear probing and MLP networks. For other baselines that modify the internal representations of the network (e.g. LoRA, VPT) InCA’s procedure can not be applied since the adaptation modifies the backbone’s execution.
>
> > The clarity of the writing can be improved. For example, it was not clear during my first pass what the dimension of z is and what T is. The description for diag-head is confusing...
>
> Thank you for the feedback, we updated the notation issues of T (# of tokens) in line 171 and on, to be consistent in our notation and remove this ambiguity. To clarify regarding, diag-linear, it is not exactly a matrix product, rather computes the diagonal of a matrix product, i.e.
>
> diag-linear([a_1, ... a_c], [w_1, ... w_c]) → [<a_1, w_1>, ... <a_c, w_c>].
>
> >The “signature” phenomenon is interesting, but the given analysis doesn’t provide much insight. It would be great to hear more about what we can learn from these “signatures,” especially given that they are presented as one of the key advantages of InCA in the conclusion.
>
> We provide an extensive analysis with insights from the signatures produced by InCA in App. B. These results were relegated to the appendix due to the page limit. Below we provide a self-contained summary of InCA’s signatures and some of the results presented in the appendix. We encourage the reviewer to check out App. B if they would to find more details about the intermediate representation signatures.
>
> + **Signatures**: Training InCA produces a set of adapters attached to different network’s activations. Evaluating each adapter (in parallel) gives the performance of each particular activation map on a task. We observe unique patterns for the signatures on different tasks indicating which intermediate representations are most helpful.
>
> + **InCA and partial tuning have matching curves (App. B.1 and Fig. 4)**
>     + In App. B.1 we compare the InCA and partial tuning signatures. In partial tuning, we repeat fine-tuning experiments where we fine-tune all layers up to a “freezing point” e.g.  training the last K layers of the network for varying values of K=1, ... L.
>
>     + In the same fashion as InCA, we produce partial tuning signature by running a set of experiments with different partial tunings and their evaluated performance. By running this for each freezing point we get a signature (albeit at a much larger cost than a single InCA training run).
>
>     + For partial tuning is that as you tune more layers - the tuning is more expressive and the test accuracy increases. Interestingly, the partial tuning plots follow an “elbow behavior” where after a certain number of layers are used for tuning, the performance improves dramatically and the performance roughly saturates afterwards (Fig. 4).
>
>     + Even more exciting, the top InCA adapter matches the location of the “elbow”. Meaning the point of the elbow is at the same location where the representation found by InCA can be harnessed by the partially tuned network.  This systematically shows the behavior of fine-tuning as “surfacing” existing representations. That is, when a particular activation layer is unfrozen it can be leveraged for downstream representation via tuning, or alternatively InCA efficiently finds those representations directly.
>
> + **Layer affinities (App. B.2, Fig. 5)**
>     + In App. B.2 we review the InCA signature of the same task applied to different backbones and architectures. (Fig. 5).
>     + What we observe is that for many datasets, the intermediate representation signature is highly consistent between different architectures and same type of pre-training to the level of exact layer matches. This intriguing property shows how much of the representations of the network are independent of the architecture and are mostly a function of the pre-training task, even for very different architectures such ViTs vs CNNs.
>
> >Applying InCA for language generation, and image diffusion
>
> Applying InCA to generative tasks is an interesting avenue for future work where we think the modularity and efficiency of InCA can be leveraged. However in the current work we focus on core discriminative tasks.
>
> We thank the reviewer for the detailed review and hope that by addressing the main concerns regarding layer selection and "signatures" that the reviewer considers raising their recommendation.

---

> > ### Comment · Reviewer_XAoM · 2023-08-14
> > **Thanks for the detailed response**
> >
> > My concerns are addressed. Please include the clarification in the revised manuscript. I've raised the score accordingly.

---

> > > ### Author Response · Authors · 2023-08-19
> > > **Response to reviewer**
> > >
> > > We are glad the rebuttal addressed the main concerns raised by the reviewer. We would like to thank the reviewer for taking the time to review the contents of our rebuttal.

---

### Official Review · Reviewer_KdhN · 2023-07-12

**Soundness:** 3 good
**Presentation:** 3 good
**Contribution:** 3 good
**Rating:** 5
**Confidence:** 4

**Summary:**

This paper proposes an efficient fine-tuning method that works parallel to the pre-trained network. Based on cross-attention between intermediate activations, InCA can generalize to various classification tasks from different domains. Additionally, the framework inherently supports class incremental learning, multi-task learning and continual learning objectives. The overall objective is cross-layer feature merging using cross-attention between them while freezing the base model.



**Strengths:**

1. There are various parameter-efficient fine-tuning methods in the literature; the major strength of this work is described in the last part of Sec. 3 - using cached activations to train the model in mere seconds.
2. The overall formulation of cross-attention is simpler and elegantly extends to multi-task and continual learning paradigms.
3. Experiments are shown on various vision classification datasets and one multi-task learning benchmark.

**Weaknesses:**

1. The formulation of Open-InCA is not completely clear and can be better presented; However, I understood the complete idea, and I had to re-read it a couple of times for deeper understanding.
2. The field of efficient fine-tuning/transfer learning is rapidly moving, and using standard benchmarks helps understand the performance gains more. The authors can give results on VTAB-1k, few-shot learning experiments, etc, as shown in SSF[1], VPT[2], FacT[3]
3. Also, I found comparison with existing methods a major weakness, as only LoRA, BiTFit and AdaLN are shown.

[1] Scaling & shifting your features: A new baseline for efficient model tuning. [2] Visual Prompt Tuning [3] FacT: Factor-Tuning for Lightweight Adaptation on Vision Transformer

**Questions:**

See the weakness sections.
1. Standard benchmarks and a thorough comparison with existing works will make this work a lot stronger
2. A major emphasis on peak training peak memory, inference latency and parameter overhead needs to be provided.

**Limitations:**

Yes

---

> ### Author Rebuttal · Authors · 2023-08-10
>
> > The formulation of Open-InCA is not completely clear and can be better presented; However, I understood the complete idea, and I had to re-read it a couple of times for deeper understanding.
>
> We thank the reviewer for the engaging review of this work and value their appreciation of the elegance of IncA for modular multi-task and CIL settings. Regarding unclear presentation of Open-InCA in Sec. 3, we first note that we provide a more extensive description and presentation of Open-InCA and class incremental learning experiments in Appendix A.
>
> As the reviewer is suggesting, we re-wrote and combined the presentation of Open-InCA in Sec. 3 that was previously presented into 2 separate parts (L178 - 184)  and continued in (L194 - 212) to give a complete picture of Open-InCA that fills in all the details directly in the main manuscript. The revised writing includes the discussion on “query only training” and highlights the disentangling of class representations for class incremental learning and forgetting.
>
> > A major emphasis on peak training peak memory, inference latency and parameter overhead needs to be provided.
>
> Regarding peak training memory we refer the reviewer to Fig. 1 (Right) which reports **peak** training GPU Memory with the training of InCA as compared with other adaptation approaches, the GPU memory is measured for the training process with a batch size of 32.
>
> Regarding inference latency, as presented in Tab. 4, the inference speed of the adapter in InCA is dwarfed by the inference of the pre-trained model, with 2.6% additional overhead for an IncA adapter on ViT L/16. This is because the InCA adapter is a small shallow network made out of a few cheap sequential operations with cross attention scaling linearly unlike self-attention. In the case of multi-task learning, the shared backbone inference computation leads to 4.4x speedup on the ImageNet to Sketch benchmark.
>
> For regular, single task inference, as noted above the overhead of InCA adapter is < 3%. Further, if the top adapter appears at a particular intermediate layer, one can discard subsequent layers and use a truncated pre-trained model.  As an example, taking ViT-L/16 with Stanf. cars (best block, 16) adaptation we run an inference benchmark for the purpose of this review and use InCA with a truncated backbone, from this experiment we observe using InCA with truncated backbone leads to 31% faster inference as compared with a fine-tuned network since InCA benefits from removing the unnecessary later layers.
>
> Lastly, regarding parameter efficiency, InCA is parameter efficient with 1.3% of the parameters of a ViT L/16.  Nonetheless as we discuss in the paper (L115-119) we believe that parameter efficiency is only a piece of the picture for an efficient adaptation which should also enable efficient training.

---

> > ### Comment · Reviewer_KdhN · 2023-08-10
> >
> > The revised writing will make things much clear. However, a major concern is a limited comparison with existing works which is still not presented in the rebuttal. This makes the work rather weak. The authors need to address this issue during the discussion period.

---

> > > ### Author Response · Authors · 2023-08-14
> > > **Response to reviewer Part 1**
> > >
> > > We disagree with the premise that we do not compare with existing work. In fact we already extensively compare with the suggested work of VPT that the reviewer ask that we compare with (already in the manuscript, e.g. see Tab. 1, Tab. 7). Below we summarize the results of VPT on ViT L/16 as compared with InCA.
> > >
> > > | Method                 | Full fine-tuning | InCA | VPT Deep |
> > > |------------------------|------------------|------|----------|
> > > | Average                | 10               | 9.8  | 12.3     |
> > > | Maximum gap to full FT | 0                | 2.3  | 6.8      |
> > >
> > >
> > > We observe that InCA has a smaller maximum gap to full-fine tuning, 2.3 vs 6.8 of VPT. In addition, in Appendix C we provide a detailed comparison with the **computational efficiency** of VPT and InCA applied to larger models. We observe that per-run, InCA is 290% faster than VPT on ViT-L/16 (with VPT efficiency issues exacerbated for bigger models, cf. line 692, Fig 1 Right). The 290% speed-up is for a single training run and the gap widens if we compare *per-dataset* due to the extensive hyper-parameter search required for good results with VPT (see the original work of VPT paper, Tab. 6).
> > >
> > > For the other works of SSF, and FaCT, we added and cite both of them as additional interesting work in the related work section for parameter efficient approaches.  Note each of SSF, FaCT and VPT require back-propagation through the entire model as we discuss (lines 75-80) this makes them expensive for fine-tuning of large-scale models akin to the computational costs of running full-fine-tuning. On the other hand InCA does not require back-propagation through the pre-trained model which makes the method much more scalable to larger models (see Fig. 1, Right for comparison of model scaling for InCA and VPT, Full-FT). Further as we discuss (lines 55 - 58) the modification of the backbone execution limits each of the approaches (SSF, FaCT, VPT) in more flexible learning and inference scenarios such as multi-task and class incremental learning, whereas InCA’s modular adaptation has a “one to many” property allowing for computation sharing of the backbone computation.
> > >
> > > In "Response to reviewer Part 2" below we compare and discuss FaCT, SSF in detail and compare them with InCA. As we note above both FaCT and SSF are parameter efficient but lack some of the other key benefits of InCA.

---

> > > > ### Author Response · Authors · 2023-08-14
> > > > **Response to reviewer Part 2: Discussion of FaCT and SSF in comparison to InCA**
> > > >
> > > > **FaCT:  FacT: Factor-Tuning** is a method enabling extreme *parameter storage* efficiency in adaptation, yet we do not believe comparing directly with FaCT is necessary for our manuscript since FaCT can be viewed akin to full-fine-tuning. The method of FaCT comprises of 2 steps: In the first step full-fine-tuning is applied to optimize the model. In the second step, the fine-tuned parameters difference is computed (Delta W), rearranged as a 3D tensor and then efficiently decomposed (e.g. via Tucker tensor decomposition). This approaches leads to immense parameter efficiency (final storage in KBs).
> > > >
> > > > The authors utilize the homogenous ViT architecture for creating a uniform 3D tensor and we note that it unclear how to use FaCT in the context of CNNs which have heterogeneous layer layout. In contrast InCA automatically applies to CNNs (see. Tab. 2).
> > > >
> > > > In the context of InCA, since in the first step of FaCT exactly applies full fine-tuning we may utilize the full-fine tuning results as a sense for the performance axis. For computational efficiency, as noted, since FaCT already contains a full fine-tuning step, it is prohibitively expensive for large-scale model (see Fig. 1 Right).
> > > >
> > > > Beyond the parameter efficiency front, FaCT suffers from all other issues of full-fine-tuning. As we discuss in the paper (lines 290-292) full-fine-tuning does not allow for fast multi-task inference presented in Tab. 4, nor class-incremental learning as shown in OpenInCA. Lastly FaCT’s adaptation results in an opaque decomposition of the parameters as compared with InCA’s performance signatures (Sec. 5, Appendix A).
> > > >
> > > > Regarding parameter efficiency, InCA uses 1.3% of the trainable parameters of a ViT-L/16 which amounts to about 4M parameters, whereas FaCT reports as low parameter count as 8K final parameters. However in the context of visual tasks - the image itself requires disk storage in the order of Megabytes hence for most applications there are diminishing returns from such extreme parameter efficiency for storage (e.g. when considering networking, IO, and storage bottlenecks). Without any storage optimizations, a single InCA adapter disk storage size is already comparable to the disk storage size of a single image. Post-training storage optimizations as presented in FaCT are complementary yet orthogonal’s to InCAs approach which can enable future storage gains if extreme storage efficiency is warranted for an application.
> > > >
> > > > **SSF:** Is an adaptation method which inserts learnable affine parameters (think LN’s affine gamma, beta) after linear operations in the network (e.g. linear operations in LN, MSHA, FF). Post-training, those affine parameters can be merged with the linear operation that preceded them since two linear operations can be composed.
> > > >
> > > > Like FaCT, SSF requires back-propagation through each of the network’s activations which makes it computationally expensive and non-scalable for large models, albeit the number of affine parameters is small which leads to parameter efficiency. Similar to FaCT, it is also *not* modular and flexible in terms of class-incremental learning, multi-task efficiency and we would expect it to have comparable computational efficiency as a method like LoRA which is 65.3% more GPU memory intensive than InCA for a ViT-H/14 model.
> > > >
> > > > We conclude that we have cited FaCT and SSF as additional context, already extensively compared with VPT in the original manuscript and provide detailed descriptions and comparisons with both FaCT and SSF above. In summary each of the above methods requires back-propagation through all of the model’s activations which makes them computationally expensive for large models, and each of the methods do not exhibit the modularity of the InCA adaptation limiting their application in the settings which we demonstrate in detail for InCA.

---

> > > > > ### Comment · Reviewer_KdhN · 2023-08-15
> > > > >
> > > > > These comments have justified the chosen methods to some extent. Still, in the case of single vision tasks, why did the authors choose the datasets in Table 1. why not the standard VTAB-1k or FGVC dataset? This is my only concern; it will make the evaluation more fair. Also, even if SSF, and FacT require backprop through the whole network, it is worth comparing them for single vision tasks and giving the training/inference memory/throughput gain to justify the claims. VTAB-1k/FGVC will make this much easier as the numbers can be directly picked from the existing works, and the authors only have to experiment with their work. A standardization of datasets/tasks is the need of the hour for efficient learning methodologies.

---

> > > > > > ### Author Response · Authors · 2023-08-19
> > > > > > **Response to Reviewer**
> > > > > >
> > > > > > We thank the reviewer for going over our response and are glad that our clarification to the questions justify the method of InCA in the eyes of the reviewer.
> > > > > >
> > > > > > With regards to training memory, we report detailed training memory comparison of different methods at different scales in Figure 1 (Right). Throughput gains are reported in Tab. 4 for multi-task inference. For inference memory, the additional required inference memory of running InCA is dwarfed compared with the memory needed for the pre-trained network inference and without any optimizations, the additional inference memory overhead of InCA is < 4%.
> > > > > >
> > > > > > With regards to the datasets, we would like to point out that in our paper we used 11 standard and official datasets which are high quality, challenging, and diverse. Each of the datasets we use is widely researched and used in the literature. The term FGVC is popularized by the CVPR FGVC workshop and we in fact consider many such FGVC datasets (see below).
> > > > > >
> > > > > > Compared with SSF for example we already use 5/6 of the datasets which SSF labels as FGVC. In particular, in Table 1 (and rest of the experiments) we already include 1) FGVC Aircrafts, 2) FGVC CUB-200, 3) FGVC Flowers (oxford flowers), 4) FGVC Cars (Stanford Cars), and 5) FGVC Dogs (Stanford Dogs). The only dataset in the FGVC consideration of SSF which is not in Tab. 1 is the FGVC NABirds dataset which considers fine-grained bird classification. Nonetheless, NABirds is correlated with the CUB-200 dataset and they both operate on the same task of fine-grained bird classification with overlap with CUB-200.
> > > > > >
> > > > > > Further in our experiments we consider additional challenging high quality datasets which are not used in SSF. For example this includes the FGVC Herbarium dataset (presented in FGVC-2020) which is considered a modern and challenging fine-grained dataset. Other high quality and diverse datasets we consider include MIT-67, Flood Depth, Describable textures dataset (DTD), EuroSAT, and Oxford Pets.
> > > > > >
> > > > > > Each of the datasets we use is an official dataset that was directly downloaded from the original source which presented the dataset (e.g. the FGVC workshops dataset website). This ensures a single source of truth at the dataset level. As compared with a dataset set like VTAB-1K, which was produced by post-processing and sub-sampling of original sourced datasets, poses challenges in reproducing index splits among other variations. For example with VTAB-1K, see this open issue [“Dataset split ids? #7”](https://github.com/google-research/task_adaptation/issues/7) concerned with not having access to consistent and direct index splits of the original datasets repository. For a dataset as small as 1000 samples, variation in which sample indices are used and the class composition lead to significant performance difference and result in additional variations and barriers for easy evaluation.

---

### Official Review · Reviewer_DnPr · 2023-07-13

**Soundness:** 3 good
**Presentation:** 3 good
**Contribution:** 3 good
**Rating:** 6
**Confidence:** 4

**Summary:**

The paper presents a method termed InCA (Introspective Cross-Attention) to learn compact adapter modules for large vision models that can be used for various downstream tasks (image classification domains). The proposed approach has an advantage over the entire model finetuning due to its parameter efficiency resulting in a smaller GPU memory footprint required for training. In addition, the proposed introspective cross-attention module is architecture-agnostic, enabling simple implementation for different intermediate network representations and network types.

**Strengths:**

The proposed approach (InCA) demonstrates stable performance when applied across different model types (ViT, SWIN, CNN). Unlike previous architecture-specific adaptors, the method can be implemented without modifications with different base model architectures.

The experiments demonstrate that the method can reach performance comparable to the entire model finetuning. At the same time, the number of trainable parameters constitutes only a tiny fraction (1-2%) of the backbone model weights. Because the training process doesn’t require backpropagation through the backbone, it is GPU memory efficient, making the large pre-trained transformer-based backbones reusable for different downstream tasks.

Compared to the linear probing, the proposed cross-attention architecture of the adaptor module has a significantly larger “extraction capacity” that leads to an improved classification accuracy of the adapted model. When applied in parallel to different activation maps, the method produces a network “signature” - w.r.t. to the downstream tasks. All this highlights the fact that the internal representations of the pre-trained large models have sufficient representation power for many downstream domains.


**Weaknesses:**

The paper builds on the large body of literature on efficient transfer learning. While exploring the utility of cross-attention as a choice for adaptor architecture, the authors follow the steps of many prior works, such as Perceiver [30] or [A]. On the other hand, the current study combines empirical results for different backbone architectures (e.g., both CNNs and Transformers) and suggests applying cross-attention modules in parallel to different activation maps to identify the most relevant features for the downstream tasks systematically. Thus, despite being a comprehensive study, the novelty is limited. As discussed in the Appendix, InCA, as an efficient and modular model adaptation framework, can be helpful for domain practitioners (e.g., medical imaging domain) to bridge the gap between cutting-edge research in visual representation learning and real-world applications.

[A] Cross-Attention is All You Need: Adapting Pretrained Transformers for Machine Translation, in EMNLP 2021.

**Questions:**

Could you highlight the most interesting (maybe counterintuitive?) findings? “The second best adaptation approach is LoRa … at additional training costs” invites the comparison along axes other than the error rate. Could you compare the training costs directly?

Table 2: “\dagger indicates full FT was avoided due to prohibitive computational costs” – could you elaborate on what costs were considered prohibitive and why?

---

> ### Author Rebuttal · Authors · 2023-08-10
>
> We thank the reviewer for the through review of our method noting the parameter/training efficiency, the signatures produced by InCA and the compactness and modularity enabled by the method as well as that InCA "can be helpful for domain practitioners". We address the points of the reviewer below.
>
> > follow the steps of many prior works, such as Perceiver [30] or [A]
>
> The reviewer is correct that we are not the first to utilize cross attention for it’s extraction capacity and indeed in the related work section we cover different uses of cross attention in the literature. We also thank the reviewer for pointing reference [A] which we have added in the related work section for it’s expressivity of cross attention. *However* regarding [A] we note that the approach by which the cross attention is used is different both in terms of the tasks considered - natural language translation, but also in 2 major ways in which the cross-attention is actually used. In the suggested work, an encoder-decoder transformer is used for machine translation task. In that setting the base architecture already crucially utilizes cross attention in the decoder layers, and the cross attention happens between the encoded tokens (as keys) and the previously generated tokens (as queries). In the work the authors show that just optimizing the cross-attention layers provides the expressivity needed for fine-tuning on the task. However we note that InCA introduces randomly initialized adapters that did not previously exist in the architecture. Further the queries themselves in the cross-attention layer used by InCA are not computed from the activations but rather are learned and optimizable parameters.
>
> Regarding the novelty of InCA, we note that while the architecture of the InCA adapter is simple, the parallel and exhaustive training of InCA does not appear in perceiver (or any other work) and is key for the efficiency and the search for relevant activations existing in the network. The lightweight and parallel approach of InCA enables it’s use to conquer large-scale pre-trained models at extremely efficient adaptation costs (e.g. adapting ViT G/14 on 1 GPU). The parallel training of InCA is also directly responsible for it’s ability to produce intermediate representation signatures in a feasible manner via the computation sharing of the different trained adapters which are all new.
>
> Further we would like to point out that we extend InCA to Open-InCA (presented in Sec. 3 and additionally in Appendix A with results) which introduces a novel adaptation approach where the learning of each class is done in a disentangled and separate fashion. This makes InCA into a highly modular and flexible adaptation approach for class-incremental learning and unlearning that further illustrates the novelty and the flexibility of the proposed approach.
>
> > “The second best adaptation approach is LoRa … at additional training costs” invites the comparison along axes other than the error rate. Could you compare the training costs directly?
>
> In our experiments we observe InCA outperforms LoRA in all architectures tested, with LoRA being the second best approach. With regard to training costs, InCA scales to larger models much more efficiently. In Figure 1 (right) we observe that since InCA does not back-propagate through the pre-trained backboned it makes training even architectures as large as ViT-Gigantic/14 feasible under fairly modest computation and GPU memory constraints. We run an experiment to compare InCA with LoRA on the a ViT-H/14 architecture, where for InCA we use 20 adapters trained in parallel. We observe that even with 20 adapters, InCA has **65.3% lower GPU memory footprint** than training one LoRA adaptation on ViT-H/14.
>
> In addition InCA and Open-InCA enable computation sharing for multi-task learning, and continual learning which are not feasible with LoRA (since LoRA modifies the backbone execution whenever it is applied to a model). Lastly InCA is architecture agnostic and can be applied to CNNs whereas LoRA dictates the existence of self-attention layers in the architecture.
>
> > dagger indicates full FT was avoided due to prohibitive computational costs” – could you elaborate on what costs were considered prohibitive and why?
>
> For each entry in Tab. 2 we run multiple training runs with different learning rate on each of the 11 datasets used in Tab. 1 to report the average accuracies on those tasks. For ViT Gigantic/14 which has 1.8B trainable parameters, running training with full-fine tuning on each of the listed datasets incurred prohibitively large costs for the given timeframe.

---

> > ### Comment · Reviewer_DnPr · 2023-08-19
> >
> > I thank the authors for their detailed response.

---

### Official Review · Reviewer_st4F · 2023-07-13

**Soundness:** 3 good
**Presentation:** 4 excellent
**Contribution:** 3 good
**Rating:** 6
**Confidence:** 4

**Summary:**

This work presents a method called Introspective Cross Attention (InCA), which aims to identify high-performing adapter models for handling downstream tasks using large-scale pre-trained models. InCA achieves competitive performance compared to the well-established baseline of full fine-tuning, while also enabling parallel multi-task inference. The experimental results effectively demonstrate the effectiveness of this method in terms of both performance and efficiency.

**Strengths:**

1. The paper is well-written and easily comprehensible. The Introduction section, in particular, effectively establishes the context for the entire work.
2. The exploration of introducing adapters for large-scale models in the context of vision transformers is still relatively less explored.
3. The promising results on well-known transfer learning datasets at a smaller scale indicate the potential of the proposed method.

**Weaknesses:**

1. The datasets employed in this study may not be the most appropriate test-bed for evaluating large-scale pre-trained models. Previous transfer learning works [1, 2, 3, 4] have achieved comparable accuracies using lightweight CNN models that demand significantly fewer FLOPs and parameters. To gain a comprehensive understanding, it would be valuable to compare the FLOPs and parameter counts.
2. While I acknowledge that the authors have utilized commonly used datasets for transfer learning tasks, it is worth noting that these datasets may not provide sufficient challenges when employing models such as ViT-G/14 and other large variants.


References
1. [Co-Tuning for Transfer Learning](https://proceedings.neurips.cc/paper/2020/hash/c8067ad1937f728f51288b3eb986afaa-Abstract.html)
2. [Stochastic Normalization](https://proceedings.neurips.cc/paper/2020/hash/bc573864331a9e42e4511de6f678aa83-Abstract.html)
3. [Bi-tuning of Pre-trained Representations](https://arxiv.org/pdf/2011.06182)
4. [$\Delta$-Networks for Efficient Model Patching](https://arxiv.org/pdf/2303.14772) -- Table-7


**Questions:**

1. Could you please provide (or comment) a comparison in terms of FLOPs?
2. In terms of memory and compute how does CNN based transfer learning approaches compare with ViT based approaches?
3. This approach might suit very well for tasks like model patching, see [PAINT](https://model-patching.github.io/) and [Delta-Networks](https://arxiv.org/abs/2303.14772)

**Limitations:**

Discussed in the appendix

---

> ### Author Rebuttal · Authors · 2023-08-10
>
> We thank the reviewer for the thoughtful review of this work and are encouraged that they find the presentation to have effective context and and the method to be be presented comprehensively including in the realm of large-scale modern architectures. We address the points raised by the reviewer below
>
> > This approach might suit very well for tasks like model patching, see PAINT and Delta-Networks
>
> We thank the reviewer for suggesting model patching. This is a very suitable testbed for the modular design of the InCA and Open-InCA adaptation, we add and cite this relevant application during our multi-task discussion in the manuscript.
>
> By not modifying the pre-trained backbone architecture and by sharing the majority of the computation with the pre-trained backbone, InCA adapters can be used for patching an existing base model. Similar to model patching are the flexible learning frameworks we consider in Sec. 3 (multi-task learning) and Appendix A (Class Incremental Learning and forgetting) which both do not exactly follow the model patching frameworks yet are highly adjacent. Namely through Tab. 4 we observe InCA is the top performing method in the ImageNet-to-Sketch multi-task benchmark and enables efficient multi-task inference which is be crucial for efficient patching networks. Further in Appendix A we show how even at the level of a single adapter and a single class one can add additional classes or unlearn a class flexibly. The corroborated results of InCA on multi-task learning and continual learning make it a great method for model patching and we added this context in the paper.
>
> > Previous transfer learning works [1, 2, 3, 4] have achieved comparable accuracies
>
> We thank the reviewer for the suggested relevant references, we have incorporated [1,2]  in the related work section as methods that focus on the learning objective and references [3,4] as additional interesting approaches for adaptation (e.g. improving normalization). We agree with the reviewer these works can be complementary to the the adaptation approach presented InCA.  However, we note when discussing similar accuracies it’s important to keep the discussion in the context of the learning task among other details. For example, the suggested work of [2] changes the *learning objective* by introducing additional un-supervised losses, as mentioned these approaches are complementary with InCA yet operate orthogonally.
>
> In general, final test accuracies may also be influenced by factors such as different data augmentations, modified image resolutions among many other boosting methodologies which improve results.  We refrain from using such boosting methods and our study simply uses the 224 input resolution with traditional random crop augmentation for fair benchmarking and to avoid conflating multiple aspects of the learning task. We can not find those details in some of the suggested work which can also make it more challenging to directly compare.
>
> > In terms of memory and compute how does CNN based transfer learning approaches compare with ViT based approaches?
>
> The memory and compute of most transfer learning methods will directly correlate with the size of the pre-trained network (especially for methods that require back-propagating through the pre-trained backbone). this is mostly independent of the architectural family between ViT and CNNs. The majority of proposed “efficient” adaptation methods are parameter efficient but are often not compute efficient (e.g. VPT, see Tab. 7 of the Appendix) which makes them challenging to be used with large scale models. On the other hand InCA is compute efficient and does not back-propagate through the pre-trained backbone making it highly scalable to massive architectures.
>
> > Could you please provide (or comment) a comparison in terms of FLOPs?
>
> InCA is computationally efficient with regards to FLOPs as compared to other methods. There is no back-propagation through the pre-trained network, which eliminates the FLOPS associated with computing gradients through each intermediate activation in the model’s operation graph. The only backwards operations being propagated are on the small adapters parameters themselves. During parallel training multiple adapters share the FLOP computations associated with the backbone's forward pass which amortizes the computation between the adapters. For a comparison in terms of training costs (measured in wall clock time) between InCA and state of the art transformer adaptation method VPT see Appendix C Tab. 7.

---

> > ### Comment · Reviewer_st4F · 2023-08-21
> >
> > I have carefully reviewed the initial submission, the authors' response, and the feedback from other reviewers. I appreciate the effort that has been invested in addressing the concerns raised, and the responses have addressed my concerns to a reasonable extent, however, I remain aligned with my initial assessment.

---

### Official Review · Reviewer_gDN4 · 2023-07-24

**Soundness:** 2 fair
**Presentation:** 2 fair
**Contribution:** 2 fair
**Rating:** 4
**Confidence:** 5

**Summary:**


Summary: Firstly, I would like to kindly point out that this paper proposes an "adaptation" method; however, I believe there may be some serious concerns in multiple aspects. Firstly, the paper appears to lack innovation in its methodology. Secondly, the experimental design also seems to have some significant shortcomings, such as lacking adequacy and detailed experimental evidence. Additionally, I would like to express my concerns about the writing aspect of the paper, where some details seem to be missing and where some over-claims are made.


**Strengths:**



(Positive) Although this paper possesses some flaws, including the lack of innovation and rigorous experimental design, I must acknowledge that the approach in this paper appears reasonable to some extent. Additionally, it is commendable that the authors have provided a significant amount of supplementary material. However, I believe these aspects may not fully compensate for the shortcomings observed in other areas of the paper. I genuinely hope that in future research, the authors will earnestly consider these critical points and endeavor to enhance the quality of their work.


**Weaknesses:**



(Negative) Allow me to further inquire about some of the claims made in the paper. In the first sentence of the article, the authors claim that in natural language tasks, the data and hypothesis space are shared, which appears quite astonishing. Considering the diversity of tasks in natural language, can this assumption be readily applicable to every task? I believe this general statement may lack sufficient basis and in-depth investigation.

(Negative) Furthermore, the authors assert that their model can enhance "robustness," but it is important to note that "robustness" is an extremely specialized concept. Therefore, I hope they can provide more detailed experimental evidence to support this claim. A simple assertion may not be sufficient to substantiate such an important proposition.

(Negative) I must emphasize that the approach presented in this paper does not seem to demonstrate significant differences compared to existing methods, including but not limited to LoRA. This apparent lack of innovation in the technical aspects is somewhat disappointing.

(Negative) Moreover, it appears that the authors' method has not undergone sufficiently large-scale model validation to demonstrate its feasibility in terms of large model transferability. Additionally, the model they have chosen may not entirely qualify as a so-called "foundation model." This lack of rigor in the experimental design raises doubts about the viability of this method.

(Negative) Additionally, the authors seem to have not thoroughly validated the capabilities of their method on various diverse tasks. The scope of their experiments appears to be rather narrow, omitting coverage of multiple tasks such as generative models and discriminative models simultaneously. For instance, in generative models, the training cost of the so-called "stable diffusion model" may be high, but its adaptation is crucial. Unfortunately, the authors' in-depth research and contributions in this area seem to be limited.


(Negative) What is also concerning is that the authors of this paper appear to have a somewhat limited perspective, primarily focusing on a few tasks they studied, while neglecting the introduction of task details. Moreover, they may hold the belief that ImageNet pretraining is the default choice worldwide, a viewpoint that is challenging to comprehend. In reality, a foundation model should not be confined to ImageNet pretraining but should consider larger benchmarks to enhance its generalization ability. I would like to strongly urge the authors to be mindful of this and clearly indicate in the paper where their model was actually pretrained.



**Questions:**


See *Weaknesses

**Limitations:**


No. See *Weaknesses

---

> ### Author Rebuttal · Authors · 2023-08-10
>
> We thank the reviewer for reviewing this work.  We address each of the reviewer’s points below:
>
> >authors hold the belief that ImageNet pre-training is the default choice worldwide, a viewpoint that is challenging to comprehend
>
> While we consider ImageNet pre-training for some of our experiments, we do not believe it to be the default choice. Indeed, we consider diverse pre-training data. In our experiments we use pre-training from different large-scale datasets **including LAION, OpenAI 400M**, and **Instagram 3.5B-17K**. Tab. 1 uses an ImageNet-22K pre-training and we note that ImageNet still serves as a strong pre-training in computer vision [1,2,3,4].
>
> >this paper does not seem to demonstrate significant differences compared to existing methods, including but not limited to LoRA. This apparent lack of innovation in the technical aspects is somewhat disappointing.
>
> As we note in lines 87-90, InCA has key differences with respect to LoRA and other common adaptation methods, which enable new use cases (continual and multi-task learning). Moreover, unlike LoRA, InCA enables massive parallelization and efficiency boost both at inference and training time (lines 55-57), with 65.3% lower GPU memory footprint than LoRA on ViT-H/14. At the same time, InCA achieves 47.7% lower relative test error than LoRA (Tab. 1). We provide the details on the differences below:
>
> * **Architecture agnostic**: InCA can be automatically extended to any backbone architecture whereas LoRA does not. InCA is simply applied on top of any architecture — including CNNs — without any change (as we show in Sec. 3, Tab. 2). On the other hand, LoRA relies on self-attention layers, limiting its use to transformers, and requires manual and cumbersome changes, replacing self-attention with new custom layers.
> * **Different learning scenarios**
>     * **Continual learning**: Unlike LoRA, the modular Open-InCA (Sec. 3) disentangles the learning of each class. This allows both class incremental learning (CIL) and unlearning (forgetting) of a class without effect on other classes. In App. A we show that Open-InCA achieves competitive results on the Split-CIFAR100 CIL benchmark.
>     * **Multi-task learning**: Unlike methods such as LoRA, VPT, etc. InCA does not alter the pre-trained model execution, therefore allowing computation sharing for multi-task inference. In Sec. 3, Tab. 4, InCA achieves the top result in terms of accuracy and performs well in inference time due to computation sharing.
> * **Computational efficiency**: InCA adapters share almost all forward computations with the backbone, however during training do not require back-propagation through it's layers. For example on a ViT-H/14 architecture, InCA with 20 learned adapters has **65.3% lower GPU memory footprint** than training one LoRA adaptation on ViT-H/14.
>
> >the authors' method has not undergone sufficiently large-scale model validation to demonstrate its feasibility..Additionally, the model they have chosen may not entirely qualify as a so-called foundation model.
>
> * Our study of InCA includes 9 separate pre-trained models each evaluated on 11 vision datasets as well as CIL and multi-task benchmarks. Our experiments include foundation models with diverse pre-trainings that have large architectures and use massive amounts of pre-training data. We provide the specific details below:
>
> * **Diverse pre-trainings:** In the paper, we experiment with models pre-trained on diverse datasets, such as
>     *  **3.5B Instagram images** based pre-training (ResNext101)[5]
>     * **400M image-text pairs** for the OpenAI CLIP pre-training (ViT-L/14)[6]
>     * **LAION 2B+ image-text pairs** used for the OpenCLIP pre-training (ViT-H/14, ViT-Gig/14)[7]
>
> * **Large architectures**: In Tab. 2 we use the ViT Gigantic/14 foundation model of OpenCLIP [7]. With over 1.8B parameters, this makes the ***largest public pre-trained vision backbone*** available today (on `timm` and `Huggingface` vision backbones).
>
> >"robustness" is an extremely specialized concept. Therefore, I hope they can provide more detailed experimental evidence to support this claim.
>
> To clarify, the paper focuses on developing a modular and efficient method generalizing to a variety of visual classification tasks. In this context, robustness of the method refers to the ability of InCA to be applied to a diverse and challenging set of classification tasks *across a variety of domains*. We added a statement in the revised paper to avoid confusion with different notions of “robustness” as defined for example in the adversarial attack or optimization communities. In Table 1, 2, 3 we observe that InCA achieves uniformly better transfer than other methods on average over the 11 different tasks and different architectures.
>
> >Allow me to further inquire about some of the claims made in the paper. In the first sentence of the article, the authors claim that in natural language tasks, “the data and hypothesis space are shared”, which appears quite astonishing.
>
> To clarify on the statement, the field of NLP has seen great progress in recent times due to LLMs. Part of this success is due to the fact that many different NLP tasks can be recasted as a sequence-to-sequence generation problem (where both the input and output sequence (data and hypothesis) live in the same shared space) [8]. For example, traditional NLP tasks such as entailment, and NER are casted as a seq-to-seq task via prompts.
> We note that paper does not aim to explore NLP foundation models and this should be taken as introductory to the narrative which then directs the reader towards the vision domain. it is **not** a main claim to be established by this work.
>
> >Using InCA in  generative models / the so-called "stable diffusion model"
>
> Our paper is concerned with performing fine-grained visual classification, a discriminative task. We do not make claims for InCA regarding generative tasks for this manuscript including stable diffusion adaptation.

---

> > ### Comment · Reviewer_gDN4 · 2023-08-22
> > **response to authors**
> >
> > -- ImageNet pretrained models are not foundation models. In this paper, only Table 2 reports a few results regarding CLIP, ViT-H/G, and ResNext-101. The primary focus of this paper is on ImageNet pretraining. Could this potentially be seen as an overclaim?
> >
> > -- If the authors are not acquainted with NLP, kindly avoid employing unscientific overclaims.
> >
> > -- If the authors are not familiar with certain scientific terms, such as "robustness," please refrain from using them without providing a precise definition.
> >
> > In summary, this paper seems to include some overclaiming, which might diminish its scientific quality. It would be advisable to consider toning down the language.

---

### Decision · Program_Chairs · 2023-09-21

**Decision:**

Accept (poster)

**Comment:**

I am recommending to **accept** this paper "Your representations are in the network: composable and parallel adaptation for large scale models".

Six reviewers provided feedback on this paper. The authors provided a response to the reviews and I appreciate the authors' comments and clarifications, specifically around the computational requirements of the presented method. There was some discussion, and two reviewers increased their scores as a result.

Among the 6 reviewers, only one reviewer's assessment falls onto the "borderline reject" side of the threshold and 3 reviewers vote for acceptance, even though only as "weak accept". Based on the review and rebuttal discussion, I would tend to put more weight on some of the reviews that evaluated the paper positively. Altogether, I think there is sufficient support among the reviewers for an acceptance of the paper.

To me personally, the method seems interesting, relatively novel in this form, well-presented, practically relevant, and the presented results are also interesting.

Overall, I believe the paper is of high enough quality to be accepted at NeurIPS, hence my recommendation to accept.